# KPF: Dominating Multi-Agent Adversarial Competition via Kalman-Inspired Policy Fusion Mechanism

## Abstract

Despite rapid advancements in Multi-Agent Reinforcement Learning (MARL), its application to complex, highly stochastic, and dynamic environments has been hindered by limitations in generalization capabilities and robustness, often resulting in degraded performance. To address this challenge, this paper proposes Kalman Policy Fusion (KPF), a novel decision fusion mechanism inspired by the Kalman filter. The core of the KPF mechanism lies in the learning-oriented adaptive weighting and iterative optimization of policy distributions during testing. Specifically, it dynamically fuses multi-agent policies while minimizing their differences, thus effectively improving generalization and robustness in highly stochastic environments and achieving exceptional performance in dynamic adversarial tasks. Furthermore, this work is the first to empirically and systematically demonstrate that the efficacy of the base models, their distributional characteristics, and their mutual complementarity are the key prerequisites that determine the upper bound of fusion performance. Comprehensive evaluations demonstrate that our mechanism establishes new SOTA benchmarks across four diverse environments, including the StarCraft Multi-Agent Challenge (SMAC) and its successor SMACv2, Google Research Football (GRF), and the Multi-Agent Particle Environment (MPE). Notably, within the complex domain of StarCraft II, KPF achieves a perfect 100% win rate on numerous challenging maps. By optimizing policy weights to approximate an unknown optimal policy, our results validate the efficacy of the Kalman-based approach in MARL decision optimization, offering valuable insights for building more robust and efficient multi-agent systems.Our code will be released on GitHub upon acceptance.

## 1 Introduction

Multi-Agent Reinforcement Learning (MARL) has demonstrated considerable potential for complex coordination tasks, including robot swarm control (Lv et al., 2025), autonomous formation flying (Huroon et al., 2025), and strategic gameplay (Wang & Pan, 2025). Nevertheless, prevailing MARL methods suffer from several fundamental limitations: non-stationarity and unstable convergence, poor sample efficiency, and limited generalization. To address these, the Centralized Training with Decentralized Execution (CTDE) framework (Lowe et al., 2017; Foerster et al., 2018) has emerged as a dominant paradigm, establishing a foundation for stable convergence by leveraging global information during training. Building on this, various communication mechanisms, such as the Transformer-based Email Mechanism (TEM) by Guo et al. (2023), have been proposed to further enhance cooperation and mitigate non-stationarity in partially observable settings.Despite these advances in stability, MARL contends with an intrinsic trade-off in its optimization paradigms. On one hand, policy gradient methods (Schulman et al., 2015; 2017; Haarnoja et al., 2018) offer theoretical convergence guarantees but are notoriously sample-inefficient. On the other hand, value decomposition methods (Foerster et al., 2018; Son et al., 2019) attempt to improve data reuse via experience replay.

However, these solutions fail to address a more fundamental bottleneck hindering the practical deployment of MARL: poor generalization. Policies are typically optimized for a specific training environment, which leads to a distributional shift that degrades performance in real-world applica-

tions. Furthermore, the highly coupled and combinatorial nature of agent-environment interactions results in learned policies that often lack robustness and generalizability when transferred to novel scenarios. In response to this challenge, prior works have explored static policy fusion (Wiering & Van Hasselt, 2008; Jia et al., 2021; Zhao et al., 2025). This approach attempts to combine multiple models through fixed rules such as voting or averaging, based on the principle of reducing variance to mitigate overfitting and thereby improve generalization. In MARL, however, this approach is often ineffective because the constituent policies frequently exhibit significant divergence—manifesting as high KL divergence in their action distributions for the same state. This high degree of disagreement can cause naive fusion methods to fail catastrophically. This ultimately highlights the inherent limitation of static mechanisms: their fixed weighting schemes lack the adaptability required for dynamic environments.

To address this issue, we propose a dynamic decision fusion mechanism inspired by Kalman filtering—Kalman Policy Fusion (KPF), which represents the first application of covariance-based online updates from state estimation theory to the final decision stage of deep reinforcement learning. This study also empirically demonstrates for the first time that *complementarity*—jointly determined by model performance and probability distributions—is a critical prerequisite for successful fusion. Evaluation on multiple benchmarks, including Google Research Football (GRF) (Kurach et al., 2020), the Multi-Agent Particle Environment (MPE) (Lowe et al., 2017), SMAC (Samvelyan et al., 2019), and SMACv2, shows that our method significantly outperforms baseline algorithms, establishing a stable and generalizable mechanism for collaborative policy optimization.

In summary, our main contributions are as follows: 1) Extensive experiments on four diverse MARL benchmarks, including SMAC and SMACv2, demonstrate that KPF achieves **100%** win rates in multiple hard SMAC scenarios and exhibits outstanding performance in the non-stationary environment of SMACv2, validating both the effectiveness and efficiency of the proposed method. 2) We explicitly identify policy diversity and complementarity as essential prerequisites for successful fusion.

## 2 RELATED WORK

In multi-agent reinforcement learning (MARL), achieving efficient cooperative decision-making is a central objective, driving continuous algorithmic improvements. Recent advances include enhancements to core paradigms, such as value function factorization (QMIX (Rashid et al., 2020; Hu et al., 2021)) and multi-agent policy gradient methods (MAPPO (Yu et al., 2022)). Concurrently, substantial research has addressed specific MARL challenges, including heterogeneity (HARL (Zhong et al., 2024)), credit assignment (GACMAC (Liu, 2024)), return uncertainty (MCMARL (Zhao et al., 2023)), and exploration (HASAC (Liu et al., 2023)), while others have focused on improving sample efficiency through techniques such as transfer learning (XfrNet (Liu et al., 2021)). However, developing new algorithms presents a fundamental challenge: the process is computationally intensive and resource-demanding, and performance often converges to a ceiling that is difficult to surpass, making further improvements elusive even for high-performing methods.

Kalman filtering (KF)(Kalman, 1960) and its variants have been widely applied in deep reinforcement learning (DRL) to address challenges ranging from physical control to algorithmic optimization. It is a state estimation method for linear Gaussian systems, capable of updating the system state in the presence of noise and uncertainty. It can be interpreted as a Bayesian-based adaptive process (Bayes, 1958), fusing prior knowledge with observations to achieve the minimum mean squared error (MMSE) optimal state estimate. In robotics and physical systems, KF is frequently employed to enhance state estimation accuracy and robustness: for example, Swift Kaufmann et al. (2023) combines visual information with KF to outperform human champions in drone racing; Joshi et al. Joshi et al. (2024) leverage KF to denoise sensors, enabling sim-to-real transfer; and Bui et al. At the algorithmic level, KF has been applied to enhance DRL performance: LKTD Shih & Liang (2024) achieves precise Q-value estimation and uncertainty modeling; UaMB-SF Malekzadeh et al. (2023) employs adaptive uncertainty estimation to guide exploration and task transfer, improving sample efficiency; Villarreal et al. Villarreal et al. (2023) use EKF to update parameter uncertainty in real time, optimizing experimental design. KalMamba (Becker et al., 2024) integrates KF inference with Mamba state-space models, achieving high efficiency and performance in long-horizon RL tasks. These works highlight the potential of combining KF with DRL to enhance policy robustness, yet

most remain confined to state or sensor estimation and have not been extended to behavioral-level fusion across multiple policies. Moreover, traditional static weighted fusion methods (Zhao et al., 2025) lack a learning-oriented mechanism, rely on fixed weight assignments, and rarely investigate intrinsic factors such as policy complementarity, often focusing solely on performance aggregation.

## 3 METHOD

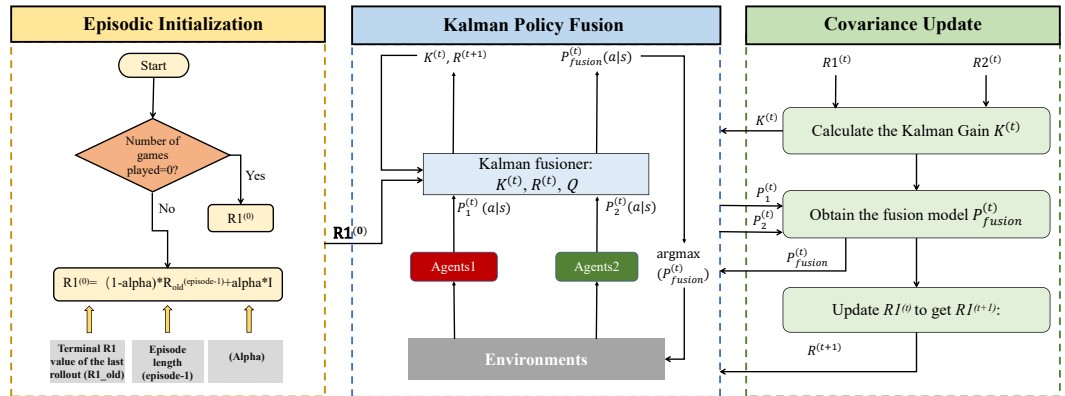

Figure 1: The Kalman Filtering Process for Fusing Policy Distributions in Multi-Agent Reinforcement Learning. The process commences with the **Episodic Initialization** stage, responsible for setting or updating the policy's uncertainty at the start of each episode. It then proceeds to the core **online policy fusion loop**, where two expert policies are combined at each timestep for decision-making. The internal computations of this loop, detailed in the **Covariance Update** stage, adhere to the principles of Kalman filtering. By iteratively computing the gain, fusing the policies, and updating the uncertainty, these stages collectively form a closed loop for continuous optimization.

In multi-agent reinforcement learning, different policy networks often produce differing action distributions for the same state, and these discrepancies may be amplified due to differences in training protocols or network architectures. A high-level diagram of our KPF mechanism is presented in Figure 1, and the algorithmic framework is detailed in Appendix A.1. Let the primary and auxiliary policies be denoted by their joint action probability distributions $\mathbf{P}_1^{(t)}(\mathbf{A}|\mathbf{S})$ and $\mathbf{P}_2^{(t)}(\mathbf{A}|\mathbf{S})$. Our objective is to construct a fused distribution $\mathbf{P}_{\text{fusion}}^{(t)}(\mathbf{A}|\mathbf{S})$ that combines the advantages of both policies to enhance robustness in decision-making. To this end, we employ principles from the Kalman filter: the primary policy $\mathbf{P}_1^{(t)}$ serves as the prior prediction, the auxiliary policy $\mathbf{P}_2^{(t)}$ as the observation, and the fused distribution is computed via uncertainty-weighted fusion. Specifically, for each agent, the primary policy $\mathbf{p}_1^{(t)}$ and the auxiliary policy $\mathbf{p}_2^{(t)}$ are represented as one-dimensional vectors, with each entry indicating the probability of choosing a specific action $a$ from the action space. In this context, $\mathbf{p}_{1,i}^{(t)}$ and $\mathbf{p}_{2,i}^{(t)}$ denote the action probability vectors of agent $i$ at time $t$. The joint action distribution of the multi-agent system can be obtained by concatenating the probability vectors of all agents, as shown in equation 1:

$$\mathbf{P}_1^{(t)} = \begin{bmatrix} \mathbf{p}_{1,1}^{(t)} & \cdots & \mathbf{p}_{1,n}^{(t)} \end{bmatrix}, \quad \mathbf{P}_2^{(t)} = \begin{bmatrix} \mathbf{p}_{2,1}^{(t)} & \cdots & \mathbf{p}_{2,n}^{(t)} \end{bmatrix} \tag{1}$$

Here, $p_{k,i}^{(a,t)}$ denotes the probability that agent $i$ selects action $a$ at time $t$ according to policy $k \in \{1, 2\}$. Each policy distribution is associated with an uncertainty covariance matrix, their structure is as shown in the following equation 2:

$$\mathbf{r}_{1,i}^{(t)} = \begin{bmatrix} m_{11}^{(t)} & \cdots & m_{1|A|}^{(t)} \\ \vdots & \ddots & \vdots \\ m_{|A|1}^{(t)} & \cdots & m_{|A||A|}^{(t)} \end{bmatrix}, \quad \mathbf{r}_{2,i}^{(t)} = \begin{bmatrix} s_{11}^{(t)} & \cdots & s_{1|A|}^{(t)} \\ \vdots & \ddots & \vdots \\ s_{|A|1}^{(t)} & \cdots & s_{|A||A|}^{(t)} \end{bmatrix} \tag{2}$$

Specifically, $\mathbf{r}_{k,i}^{(t)}$ denotes the covariance matrix of agent $i$ in policy $k$ at time $t$, which is a $|A| \times |A|$ matrix characterizing the estimation uncertainty, where $|A|$ represents the size of the action space.

The diagonal element $m_{aa}^{(t)}$ represents the variance of the probability estimate for action $a$, with smaller values indicating higher confidence.

The element-wise computation of the Kalman gain matrix $\mathbf{k}$ is given in equation 3, which serves as the core weighting mechanism in the fusion process.

$$\mathbf{k}_i^{(t)} = \mathbf{r}_{1,i}^{(t)} \left( \mathbf{r}_{1,i}^{(t)} + \mathbf{r}_{2,i}^{(t)} + \epsilon \mathbf{I} \right)^{-1} \tag{3}$$

Here, $\mathbf{r}_{1,i}^{(t)}$ and $\mathbf{r}_{2,i}^{(t)}$ are the covariance matrices of the primary and auxiliary policies for agent $i$ (as in equation 2), with $\epsilon \mathbf{I}$ added for numerical stability ($\epsilon = 1 \times 10^{-6}$, $\mathbf{I}$ is the identity matrix). The gain dynamically regulates the incorporation of corrections from the auxiliary policy based on the relative uncertainties, and policy fusion is performed as in equation 4:

$$\mathbf{p}_{f,i}^{(t)} = \mathbf{p}_{1,i}^{(t)} + \mathbf{k}_i^{(t)}(\mathbf{p}_{2,i}^{(t)} - \mathbf{p}_{1,i}^{(t)}) \tag{4}$$

Where $\mathbf{p}_{1,i}^{(t)}$ and $\mathbf{p}_{2,i}^{(t)}$ are the primary and auxiliary policy vectors (as in equation 1), $\mathbf{p}_{f,i}^{(t)}$ is the fused policy, and $\mathbf{k}_i^{(t)}$ the Kalman gain (computed in equation 3).

Subsequently, the covariance matrix of the primary policy is updated using the Joseph form, as detailed in equation 5:

$$\mathbf{r}_{1,i}^{(t+1)} = (\mathbf{I} - \mathbf{k}_i^{(t)})\mathbf{r}_{1,i}^{(t)}(\mathbf{I} - \mathbf{k}_i^{(t)}) + \mathbf{k}_i^{(t)}\mathbf{r}_{2,i}^{(t)}\mathbf{k}_i^{(t)} \tag{5}$$

Where in, $\mathbf{r}_{1,i}^{(t+1)}$ denotes the primary policy covariance matrix of agent $i$ at time $t + 1$. Here, the gain matrix $\mathbf{k}_i^{(t)}$ is obtained from equation 3, while the prior covariance matrices $\mathbf{r}_{1,i}^{(t)}$ and $\mathbf{r}_{2,i}^{(t)}$ are defined in equation 2. To preserve the geometric properties of the covariance matrix, an explicit symmetrization is applied to each element, as shown in equation 7:

$$\mathbf{r}_{1,i}^{(t+1)} \leftarrow \frac{1}{2} \left( \mathbf{r}_{1,i}^{(t+1)} + \mathbf{r}_{1,i}^{(t+1)} \right) \tag{6}$$

To refine the covariance matrix of the primary policy and achieve improved performance (as will be demonstrated in the experimental section), a memory decay mechanism is applied, as shown in equation 7:

$$\mathbf{r}_{1,i}^{(0)} = (1 - \alpha)\mathbf{r}_{1,i}^{\text{old}} + \alpha * \mathbf{r}_{1,\text{init}} \tag{7}$$

In this formulation, $\mathbf{r}_{1,i}^{(0)}$ denotes the primary policy covariance matrix of agent $i$ at time $t = 0$, while $\mathbf{r}_{1,i}^{\text{old}}$ corresponds to the final-step covariance matrix of agent $i$ from the previous iteration. The scalar $\alpha$ represents the memory decay factor, and $\mathbf{r}_{1,\text{init}}$ denotes the initial covariance, initialized as the identity $I$.

To ensure numerical stability and guarantee that the fused vector $\mathbf{p}_f^{(t)}$ constitutes a valid probability distribution, it is subjected to clipping and normalization, as illustrated in equation 8:

$$p_{f,i}^{(a,t)} = \frac{\text{clip}\left(p_{f,i}^{(a,t)}, \delta_{\min}, \delta_{\max}\right)}{\sum_{a' \in \mathcal{A}} \text{clip}\left(p_{f,i}^{(a',t)}, \delta_{\min}, \delta_{\max}\right)}, \quad \forall a \in \mathcal{A} \tag{8}$$

Where in, $p_{f,i}^{(a,t)}$ denotes the probability of action $a$ for agent $i$ at time step $t$ according to the fused policy. The function $\text{clip}(\cdot, \delta_{\min}, \delta_{\max})$ restricts the probability values within the range $[\delta_{\min}, \delta_{\max}]$, preventing excessively small or large probabilities. The iterative process in the above equation is designed to adjust the Kalman gain $\mathbf{K}$, enabling the fused policy to progressively approach the unknown optimal policy distribution, as shown in equation 9:

$$\min_{\mathbf{k}} \mathbb{E}[\|\mathbf{p}_{\text{true}}^{(t)} - \mathbf{p}_{f,i}^{(t)}] \tag{9}$$

Where $\mathbf{p}_{\text{true}}$ denotes the unknown optimal policy distribution. The Kalman gain $\mathbf{k}$ is designed to minimize the trace of the posterior covariance, ensuring that the fused policy $\mathbf{p}_{\text{fusion}}$ constitutes the Best Linear Unbiased Estimator (BLUE).

Theoretically, the equations above collectively ensure the robustness and convergence of the mechanism. As the system progressively integrates valid information, the primary policy's covariance matrix gradually decreases, indicating a reduction in uncertainty and an increase in decision-making confidence.

# 4 EXPERIMENTS

To comprehensively validate the effectiveness of the proposed Kalman fusion mechanism, we assessed its environmental adaptability across multiple reinforcement learning benchmarks, examined its generalization through cross-model fusion experiments, and revealed a positive correlation between weight updates and performance via ablation studies on the number of Kalman iteration steps, thereby confirming the method's efficacy and enhancing its interpretability.

## 4.1 EXPERIMENTAL SETTINGS

**Environments.** This study selects four multi-agent reinforcement learning test environments—SMAC(Samvelyan et al., 2019), SMACv2, GRF(Kurach et al., 2020), and MPE(Lowe et al., 2017)—to establish a multi-tiered evaluation framework. Each environment presents distinct challenges: MPE, as a lightweight particle world, facilitates the validation of fundamental coordination and communication skills; SMAC provides a standardized tactical micromanagement benchmark in discrete action spaces; SMACv2 introduces high stochasticity to evaluate the algorithm's generalization and robustness in non-stationary environments; and GRF, based on continuous control and football rules, assesses agents' capacity for advanced strategy and teamwork in complex scenarios.

**Baselines.** We evaluate the proposed fusion method by benchmarking it against state-of-the-art multi-agent reinforcement learning (MARL) algorithms, namely QMIX (Rashid et al., 2020), MAPPO (Yu et al., 2022), HATRPO (Zhong et al., 2024), and HASAC (Liu et al., 2023). To ensure fair comparison, all baseline algorithms strictly follow the hyperparameter settings recommended in their original publications. During training, each model is fully trained until it consistently achieves a high win rate on its respective map, ensuring that the baseline models integrated into the fusion framework are well-converged and represent high-performance architectures, thereby providing a solid foundation for subsequent fusion studies.

**Evaluation Metrics.** To guarantee a thorough and equitable assessment, we utilize well - established, domain - relevant performance metrics tailored to each benchmark environment.The reported performance metrics for each task, namely the mean win rate and its standard deviation, are aggregated across a minimum of five distinct random seeds. For each individual seed, the reported win rate is itself the average outcome of a minimum of 100 independent evaluation episodes. In the StarCraft II suite (SMAC/SMACv2), we use the win rate as the primary metric for evaluating performance. For the Google Research Football (GRF) environment, performance is assessed using the scoring rate, as it directly reflects the agent's scoring capability. For the Multi-Agent Particle Environment (MPE), which typically involves cooperative tasks, we adopt the average collective episodic reward as the key performance indicator, effectively capturing the team's overall success in achieving shared objectives.

## 4.2 RESULTS AND ANALYSIS

Experimental results demonstrate that the proposed Kalman Fusion Mechanism (KPF), by effectively integrating HAPPO and HATRPO, exhibits superior performance in the SMAC environment. As shown in Table 1, KPF achieves state-of-the-art (SOTA) performance on 8 out of 9 evaluated maps. Notably, in the Hard scenarios, KPF achieves a perfect 100% win rate on three of the four test maps, implying an undefeated record across over 1,500 matches. Even on the 5m_vs_6m map, its win rate remains robust between 95% and 99%, substantially outperforming other methods.In the Super Hard scenarios, KPF's superiority remains pronounced: while baseline methods exceeded a 95% win rate in only 12% of cases, KPF achieved this level in 80% of cases; furthermore, no baseline method surpassed a 98% win rate, a threshold KPF exceeded in 60% of scenarios, representing a substantial performance gain on these challenging tasks. This dominant performance stems from the framework's strong synergistic effect and robustness. For instance, in the 3sv5z_vs_3sv6z scenario, KPF fuses two policies with individual win rates of 66.2% and 72.5% to create a superior strategy achieving a 91.6% win rate. Under a specific random seed, even with a 21-percentage-point performance gap (60% vs 81%, as detailed for 3sv5z_vs_3sv6z in Table 4) between policies, KPF achieves an even higher 95% win rate, demonstrating its core value in dynamically leveraging complementary strengths.

| Map | KPF | HASAC | HAPPO | HATRPO | MAPPO | QMIX |
|---|---|---|---|---|---|---|
| **Evirment:** SMAC | | | **Difficulty:** Hard | | | |
| 8m_vs_9m | **100(0.0)** | 97.5(1.2) | 83.8(1.4) | 92.5(3.7) | 87.5(4.0) | 92.2(1.0) |
| 5m_vs_6m | **96.4(1.4)** | 90.0(3.9) | 77.5(7.2) | 75.0(6.5) | 75.0(18.2) | 77.3(3.3) |
| 3s5z | **100(0.0)** | **100(0.0)** | 97.5(1.2) | 93.8(1.2) | 96.9(0.7) | 89.8(2.5) |
| 10m_vs_11m | **100(0.0)** | 95.0(3.1) | 87.5(6.7) | 98.8(0.6) | 96.9(4.8) | 95.3(2.2) |
| **Evirment:** SMAC | | | **Difficulty:**Super Hard | | | |
| MMM2 | 97.2(1.2) | **97.5(2.4)** | 88.8(2.0) | 97.5(6.4) | 93.8(4.7) | 87.5(2.5) |
| 3sv5z_vs_3sv6z | **91.6(2.7)** | 82.5(4.1) | 66.2(3.1) | 72.5(14.7) | 70.0(10.7) | 87.5(12.6) |
| 27m_vs_30m | **99.2(0.5)** | - | 76.6(1.3) | 93.8(2.1) | 80.0(6.2) | 45.3(14.0) |
| corridor | **99.4(0.6)** | 90.0(10.8) | 92.5(13.9) | 88.8(2.7) | 97.5(1.2) | 82.8(4.4) |
| 6h_vs_8z | **100(0.0)** | 95.0(3.1) | 76.2(3.1) | 78.8(0.6) | 85.0(2.0) | 92.2(26.2) |
| **Evirment:** SMACv2 | | | **Difficulty:** - | | | |
| protoss_5_vs_5 | **72.2(1.3)** | - | 57.5(1.2) | 50.0(2.4) | 56.2(3.2) | 65.6(3.9) |
| terran_5_vs_5 | **74.6(0.9)** | - | 57.5(1.3) | 56.8(2.9) | 53.1(2.7) | 62.5(3.8) |
| zerg_5_vs_5 | **63.6(1.7)** | - | 42.5(2.5) | 43.8(1.2) | 40.6(7.0) | 34.4(2.2) |
| zerg_10_vs_10 | **50.8(1.5)** | - | 28.4(2.2) | 34.6(0.2) | 37.5(3.2) | 40.6(3.4) |
| zerg_10_vs_11 | **41.2(0.9)** | - | 16.2(0.6) | 19.3(2.1) | 29.7(3.8) | 25.0(3.9) |

Table 1: The average test win rates (%) on SMAC and SMACv2 benchmarks. KPF was evaluated against a set of strong baseline algorithms. The table shows mean test win rates and standard deviations over five evaluation seeds, with the highest values highlighted in bold.

In the SMACv2 benchmark, the KPF framework demonstrates significant performance improvements, primarily due to its ability to effectively handle severe non-stationarity introduced by dynamic AI opponents (As detailed in the last five rows of Table 1). In this environment, conventional single-agent policies are easily countered, whereas KPF achieves an average absolute win-rate improvement of over 17 percentage points compared to the stronger of its two expert policies (HAPPO and HATRPO). This advantage arises from its unique online dynamic weighting mechanism: KPF acts as an online arbiter, assessing the uncertainty of constituent policies at each timestep using its Kalman-based core, and adaptively increasing the influence of the more reliable expert, thereby forming more stable and coherent team-wide strategies. The data reveal a synergistic effect, in which the combination outperforms the sum of its components: on the `zerg 10 vs 11` map, KPF achieves a final win rate of 41.2%, more than double that of its constituent expert policies. Therefore, KPF's outstanding performance on SMACv2 indicates that for complex, non-stationary adversarial problems, shifting from learning monolithic policies to constructing adaptive mechanisms that dynamically integrate multiple experts represents an effective pathway to achieving greater robustness and scalability.

In contrast to the significant gains observed in SMAC, KPF exhibits more subtle behavior in the MPE. While its overall performance aligns closely with the baseline algorithms, it consistently achieves marginal yet stable improvements (Table 2), acting as a fine-tuner that extracts residual gains from subtle differences in policy distributions. For instance, in the **Spread_v2** environment, the best-case improvement over MAPPO is **0.1863**, yet this gain remains relatively modest. The limited gains stem from the "over-confident" nature of the baseline policies in MPE—their probability distributions are overly sharp, concentrating mass on nearly a single action (a detailed discussion and analysis is provided in Section A.3.1 of Appendix A.3). This provides minimal complementary information and thus constrains the potential for synergistic improvement. The effectiveness of model complementarity relies on two key conditions: the performance gap between models should not be too large, and their predictive distributions must exhibit sufficient diversity. When one model can compensate for another in certain situations, the fusion effect is strengthened. Specifically, the inherent diversity of base policies provides richer information and thereby increases the success rate of fusion, while a smaller performance gap prevents weak models from introducing disruptive errors due to a lack of complementary value. This analysis highlights a critical insight: research on model fusion should not only focus on **how** to fuse models but also on **which** models ought to be fused.

Based on our analysis in the MPE environment, we identify the key prerequisites for effective policy fusion. The Google Research Football (GRF) environment provides an ideal testbed for this investigation. Even state-of-the-art strategies retain significant probabilities on suboptimal actions, which

we hypothesize contain latent complementary information critical for KPF's synergistic potential. However, initial attempts to fuse expert models yielded limited effectiveness. To test this hypothesis, we designed two preprocessing methods: Top-K confidence alignment (smoothing overconfident policies) and Top-K intersection filtering (retaining only mutually endorsed actions). Although these approaches improved performance, they remained essentially symptomatic solutions. To fundamentally address the problem, we adopted a systematic co-design approach, retraining the base expert models to make them explicitly fusion-ready. While maintaining SOTA-level performance, the retraining encourages smoother policy distributions, ensuring each expert retains its expertise while remaining responsive to corrective signals from its partners. Implementation details for the co-design method, including the two preprocessing approaches and the expert model retraining process, are provided in the Appendix A.3.2.

Table 3 validates this approach: in the "3v.1 with keeper" scenario, KPF achieved a 98.2% win rate, and in the coordination-intensive "RPS" scenario, it reached 82.2%, substantially surpassing the performance of individual experts, with other scenarios also showing notable improvements. The success in GRF arises not only from a powerful fusion algorithm but also from a comprehensive co-design philosophy: identifying prerequisites for fusion, diagnosing and refining base models, and achieving performance gains through joint optimization. This provides a key insight for the multi-agent ensemble domain: the most effective systems emerge from the co-evolution and deep adaptation of expert components and the fusion mechanism itself.

Our experiments reveal a key prerequisite for enhancing synergy: base models must exhibit policy diversity. This implies that policy distributions should not be overly sharp and must retain suboptimal actions beyond the optimal one, providing exploitable complementary information for fusion. Our study shows that KPF can robustly integrate information and synthesize a superior policy even when base models exhibit disparate performance, all while preventing performance degradation. In multi-agent environments, each timestep offers KPF multiple opportunities for stable policy refinement, and the accumulation of these adjustments ultimately yields substantial overall performance gains, explaining KPF's outstanding results in environments such as StarCraft.

| Scenario | KPF | HAPPO | HATRPO | MAPPO |
|---|---|---|---|---|
| **Evirment:** MPE | | | **Action Space:** Discrete | |
| Reference_v2 | **-13.6111(0.3229)** | -14.5993(0.2744) | - | -13.6146(0.3225) |
| Speaker_v3 | **-11.0034(1.2335)** | -11.0273(1.2252) | -11.0103(1.2366) | - |
| Spread_v2 | **-59.1665(0.8361)** | - | -65.0284(1.2440) | -59.3528(0.9234) |

Table 2: Performance comparisons in the Multi-Agent Particle Environment (MPE), reporting mean episode rewards and their corresponding standard deviations. For brevity, we use abbreviated names for the scenarios; their full descriptions are provided in the Appendix A.3.

| Scenario | KPF | HAPPO | MAPPO | QMIX |
|---|---|---|---|---|
| **Evirment:** GRF | | **Difficulty:** - | | |
| 3v.1 | **98.2(0.8)** | 94.74(3.05) | 88.03(4.15) | 8.12(4.46) |
| CA(easy) | **96.8(1.7)** | 92.00(1.62) | 87.76(6.40) | 15.98(11.77) |
| CA(hard) | **93.8(2.3)** | 88.14(5.77) | 77.38(10.95) | 3.22(4.39) |
| RPS | **82.2(1.8)** | 77.30(7.40) | 76.83(3.57) | 8.08(3.29) |

Table 3: Performance comparisons in the Google Research Football (GRF) environment, showing mean test scoring rate (%) and standard deviations across four different scenarios. The scenario names herein are also abbreviated for brevity; full descriptions are provided in the AppendixAppendix A.3.

## 4.3 ABLATION STUDY

### 4.3.1 ANALYSIS OF KEY HYPERPARAMETERS

The key hyperparameters of KPF ($r2$, $\alpha$, noise_level) are tuned based on theoretical priors rather than blind search, which substantially reduces computational overhead. The relative confidence parameter $r2$ is established from the win-rate ratio of the two baseline models, often yielding

| Map | Seed | Alpha | $r_2$ | Noise Level | Win Rate (%) |
|---|---|---|---|---|---|
| 10m_vs_11m | 2 | 0.0 | 0.97/0.80 | 0.000 | 97.0 |
| 10m_vs_11m | 2 | 0.0 | 0.97/0.80 | 0.007 | 99.0 |
| 10m_vs_11m | 2 | 0.0 | 0.97/0.80 | 0.008 | 100.0 |
| 27m_vs_30m | 1 | 0.0 | 0.94/0.78 | 0.0 | 96.0 |
| 27m_vs_30m | 1 | 0.4 | 0.94/0.78 | 0.0 | 100.0 |
| 3s5z_vs_3s6z | 5 | 0.0 | 0.81/0.60 | 0.000 | 75.0 |
| 3s5z_vs_3s6z | 5 | 0.3 | 0.81/0.60 | 0.000 | 87.0 |
| 3s5z_vs_3s6z | 5 | 0.30 | 0.81/0.60 | 0.001 | 95.0 |

Table 4: Ablation study on the sensitivity of key hyperparameters across different maps and seeds. Further details are provided in Appendix A.4, as specifically detailed in Table 7.

excellent initial performance. A targeted linear search within a small subsequent interval (e.g., $\pm 0.05$) is then sufficient to identify the optimal configuration. For the inter-episode memory parameter $\alpha$, full knowledge transfer ($\alpha = 0$) serves as a strong baseline, naturally constraining the search to a small, effective range. The process noise `noise_level` primarily functions as a fine-tuning parameter for pushing performance from near-optimal to optimal. This prior-guided tuning strategy enables the rapid identification of optimal configurations at minimal computational cost, as demonstrated by the results in Table 4. For instance, on the `10m_vs_11m` map, with other parameters fixed, solely adjusting the `noise_level` pushes the win rate from an already high 99.0% (at 0.007) to the optimal 100.0% (at 0.008). Similarly, on the `27m_vs_30m` map, starting from the strong baseline of $\alpha = 0.0$ (96.0% win rate), a targeted search finds the optimal configuration at $\alpha = 0.4$. The results for `3s5z_vs_3s6z` further validate our staged strategy: first, a superior win rate of 87.0% is achieved by tuning $\alpha$ to 0.30 (with `noise_level` at 0); subsequently, fixing $\alpha = 0.30$ and fine-tuning the `noise_level` boosts performance to 95.0%. Collectively, these results confirm that our prior-guided approach efficiently identifies high-performing hyperparameter configurations.

### 4.3.2 CROSS-MODEL FUSION

| Algorithm | Difficulty: Hard | | Difficulty: Super Hard |
|---|---|---|---|
| | 8m_vs_9m | 3s5z | 6h_vs_8z |
| HAPPO+MAPPO | 96.6(0.9) | 99.8(0.4) | 98.0(1.2) |
| HATRPO+MAPPO | 99.6(0.5) | 99.4(0.5) | **100(0.0)** |
| HAPPO+HATRPO | **100(0.0)** | **100(0.0)** | **100(0.0)** |
| HAPPO | 83.8(1.4) | 97.5(1.2) | 76.2(3.1) |
| HATRPO | 92.5(3.7) | 93.8(1.2) | 78.8(0.6) |
| MAPPO | 87.5(4.0) | 96.9(0.7) | 85.0(2.0) |

Table 5: Examines the effectiveness of cross-model policy fusion. It presents performance comparisons across three SMAC maps, showing results when our method fuses different algorithmic models with base policies. Reported metrics include mean test win rates (%) and standard deviations.

A key question is whether the synergistic effect of KPF is limited to a specific pairing of agents. To investigate this and comprehensively evaluate the decision quality and generalization capability of our fusion mechanism, we conducted a series of plug-and-play cross-model policy fusion experiments that require no additional fine-tuning. As shown in Table 5, the results clearly demonstrate that fusing different model pairs yields final policies that consistently outperform their respective baseline models. For instance, `HAPPO+HATRPO` achieved a 100% win rate across all test scenarios, `HATRPO+MAPPO` averaged 99.7% across three scenarios, and `HAPPO+MAPPO` reached 98.0% on the `6h_vs_8z` map, highlighting the mechanism's effectiveness and generality. These findings provide strong evidence for the cross-model generalization of our method and refute concerns that KPF is limited to specific algorithms.

### 4.3.3 ABLATION ON UPDATE COUNT K

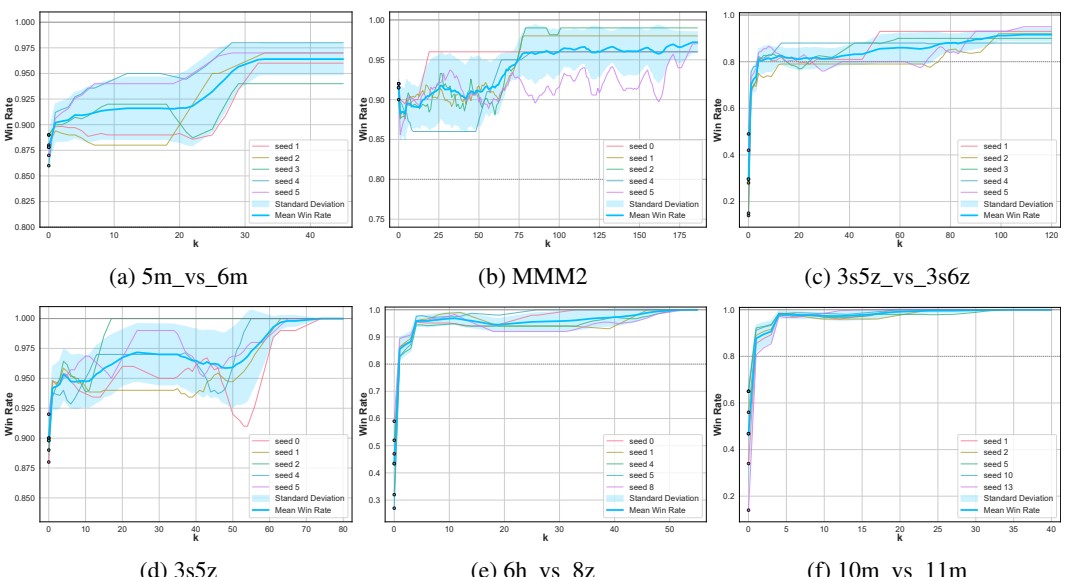

Figure 2: The effect of fusion steps ($k$) on performance across different SMAC maps is analyzed. The figure shows mean win rates (blue solid line) and standard deviations (shaded area) as a function of the number of fusion iterations $k$, based on multiple random seeds.

Our ablation study on the KPF update step, $k$, reveals its intrinsic, multi-stage learning dynamics. As shown in Figure 2, the results across all nine scenarios underscore the importance of online adaptation, as even brief adaptation markedly outperforms static fusion ($k = 0$).(plots for the remaining scenarios are provided in Appendix A.2, specifically in Figure 3) The learning characteristics vary with scenario complexity: in maps like `10m_vs_11m` and `6h_vs_8z`, convergence is rapid and stable; whereas in `5m_vs_6m` and `3s5z_vs_3s6z`, non-monotonic dynamics emerge, including performance dips and requiring a longer adaptation period to achieve high, stable win rates. On a more challenging map like `MMM2`, this process is more protracted and unstable, exhibiting significant fluctuations as $k$ increases before ultimately reaching the desired win rate. The eventual stabilization of performance across all scenarios, as shown in Figure 2, reflects the convergence properties of the Kalman filter. This behavior corresponds to the filter's underlying mechanics: as uncertainty decreases, the Kalman gain is reduced, causing the system to smoothly transition from a high-gain, exploration-focused "learning" mode to a low-gain, exploitation-focused "execution" mode. Therefore, $k$ should not be viewed as a simple hyperparameter but as a key regulator of the online learning dynamics. Its optimal value represents a critical trade-off between adaptation and stable convergence. This reveals a core insight: in multi-agent systems, the most effective strategy emerges not from perpetual adaptation, but from consolidating learning gains at the opportune moment.

## 5 CONCLUSIONS

In conclusion, this work highlights the effectiveness of dynamic, learning-oriented ensemble methods in multi-agent decision-making. By analyzing the limitations of existing approaches, we identify a key prerequisite for successful fusion—the complementarity of base models. To address the generalization problem in reinforcement learning, we propose KPF, a novel mechanism inspired by the Kalman filter, which for the first time applies dynamic systems theory at the behavioral decision fusion layer to adapt to environmental stochasticity. Extensive experiments demonstrate that KPF consistently and significantly outperforms state-of-the-art baselines across multiple challenging benchmarks, showcasing its robustness in non-stationary environments. Our work not only advances the understanding of multi-agent ensemble methods but also provides a practical and efficient fusion mechanism, offering valuable insights for the application of control-theoretic tools in the policy fusion domain.

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

# A APPENDIX

## A.1 ALGORITHM AND IMPLEMENTATION DETAILS

Theoretically, this set of equations ensures the robustness and convergence of the KPF mechanism. As the system progressively integrates valid information, the primary policy's covariance, $\mathbf{R}$, gradually decreases, signifying a reduction in uncertainty and a corresponding increase in decision-making confidence. When $\mathbf{R}$ converges to a stable minimum, the Kalman gain $\mathbf{K}$ also stabilizes, leading the system to a convergent state. Furthermore, the mechanism achieves mathematical optimality via Minimum Mean Squared Error (MSE) estimation. In this formulation, the covariance matrix $\mathbf{R}$ effectively represents the MSE of the policy estimate, and its trace, $\mathrm{tr}(\mathbf{R})$, directly quantifies the total MSE. Table 6 presents the pseudocode for the Kalman-Ensemble Policy Fusion (KPF) method.

| Step | Operation |
|---|---|
| **Inputs** | |
| | - Heterogeneous policies: $\{\pi_1, \pi_2, ..., \pi_M\}$ |
| | - Environment $E$ with state space $S$ and action space $A$ |
| | - Initial covariance matrix $\Sigma_{\text{init}} \in \mathbb{R}^{|A| \times |A|}$ |
| | - Base noise covariance $\Sigma_0 \in \mathbb{R}^{|A| \times |A|}$ |
| | - Hyperparameters: covariance mixing coefficient $\alpha$, clipping threshold $\gamma$, max episodes $T_{\text{max}}$ |
| | - Kalman regularizer $\lambda = 10^{-6}$ |
| **Initialize** | |
| | 1. Load pretrained models $F_\omega$ and $F_{1,\omega_1}$ for policy functions. |
| | 2. For all agents $k \in \{1, ..., N\}$: set $\Sigma_k \leftarrow \Sigma_{\text{init}}$ |
| **Main Loop (Episodes $t = 1$ to $T_{\text{max}}$)** | |
| | 1. Reset environment: $o_0, s_0 \leftarrow E.\text{reset}()$ |
| | 2. Time step loop ($m = 1$ to $M$): |
| | For all agents $k \in \{1, ..., N\}$ in parallel: |
| | $\quad P_1^{(k)} \leftarrow F_\omega(o^{(k)}, s^{(k)})$ |
| | $\quad P_2^{(k)} \leftarrow F_{1,\omega_1}(o^{(k)}, s^{(k)})$ |
| | $\quad K_k \leftarrow \Sigma_k(\Sigma_k + \Sigma_0 + \lambda I)^{-1}$ |
| | $\quad P_{\text{fusion}}^{(k)} \leftarrow P_1^{(k)} + K_k(P_2^{(k)} - P_1^{(k)})$ |
| | $\quad a_t^{(k)} \leftarrow \mathrm{argmax}(\mathrm{clip}(P_{\text{fusion}}^{(k)}, \gamma, 1 - \gamma))$ |
| | $\quad \Sigma_k \leftarrow (I - K_k)\Sigma_k(I - K_k)^\top + K_k\Sigma_0 K_k^\top$ |
| | $\quad \Sigma_k \leftarrow 0.5 \times (\Sigma_k + \Sigma_k^T)$ |
| | Execute joint action $a_t$, observe reward $r_t$ and next state $s_{t+1}$ |
| | 3. After time steps, update covariance for all agents $k \in \{1, ..., N\}$: $\Sigma_k \leftarrow (1 - \alpha)\Sigma_k + \alpha\Sigma_{\text{init}}$ |

Table 6: Kalman-Ensemble Policy Fusion (KPF)

## A.2 SUPPLEMENTARY RESULTS FOR THE ABLATION STUDY ON $k$

To further supplement the ablation study on the KPF update step, $k$, this section provides the learning dynamics curves for additional SMAC scenarios. Figure 3presents detailed results for the 8m_vs_9m, 27m_vs_30m, and corridor maps(Details for each scenario are provided in Figure 3a, Figure 3b, Figure 3c). In each subplot, the test win rate is plotted as a function of the number of fusion steps, $k$. The plots show individual training curves for multiple random seeds (thin lines), the mean win rate (thick blue line), and the standard deviation (shaded region). These supplementary results are consistent with the findings in the main text, reaffirming that our method converges to a high win rate and subsequently maintains stability, thereby demonstrating its robustness across different scenarios.

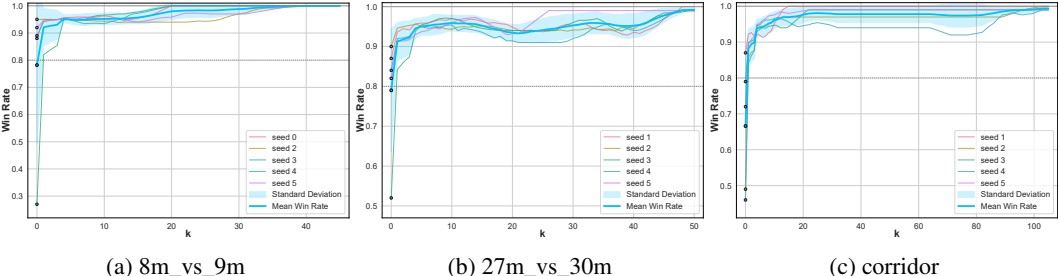

(a) 8m_vs_9m        (b) 27m_vs_30m        (c) corridor

Figure 3: Supplementary analysis of the impact of fusion steps $k$ on performance in other SMAC scenarios. Each subplot shows the learning curves for multiple random seeds, the mean win rate (thick blue line), and the standard deviation (shaded area). The results show that KPF exhibits similar rapid convergence and robust insensitivity to the value of $k$ in these supplementary scenarios.

### A.3 ON THE PREREQUISITES FOR EFFECTIVE POLICY FUSION

In our experiments, we evaluate agents across two widely used MARL environments, namely the **Multi-Agent Particle Environment (MPE)** and **Google Research Football (GRF)**. Within MPE, we adopt the commonly used scenarios: *simple_reference_v2-discrete* (abbreviated as **Reference_v2**), *simple_speaker_listener_v3-discrete* (**Speaker_v3**), and *simple_spread_v2-discrete* (**Spread_v2**). For GRF, we consider diverse tasks that vary in difficulty and coordination requirements, including *academy_3_vs_1_with_keeper* (**3v1**), *academy_counterattack_easy* (**CA (easy)**), *academy_counterattack_hard* (**CA (hard)**), and *academy_run_pass_and_shoot_with_keeper* (**RPS**).

#### A.3.1 MPE ENVIRONMENT

**To illustrate the "limited gains stemming from the over-confident nature of baseline policies in MPE" mentioned in the main text**, we provide a detailed analysis below on the extreme characteristics of the baseline policy distributions and their impact on the fusion performance. In the **Multi-Agent Particle Environment (MPE)**, the policies of the initially used baseline models (_old) exhibit extremized characteristics. Specifically, HAPPO's policy distribution is **highly peaky**, concentrating nearly all probability mass on a single action, with other actions having near-zero probability. While MAPPO's distribution is slightly smoother, its optimal action's probability remains significantly higher than others, making it difficult for its suboptimal actions to provide effective complementary information. This effectively collapses the operational space for distributional fusion from the outset (As shown in Figure 4a, 4c).

Consequently, as HAPPO's output is almost entirely focused on a single action, it fails to provide valuable corrective signals for MAPPO. The fusion algorithm is thus limited to a trivial combination between HAPPO's optimal action and MAPPO's distribution, without the ability to leverage complementarity from suboptimal actions. Further analysis reveals that even when HAPPO's primary action is correct, its extreme distribution can adversely skew MAPPO's estimates on low-probability actions, leading to performance degradation. This issue persists even when employing Top-K strategies. A small K value fails due to a lack of intersection between the policies, while a large K value introduces excessive noise, further degrading performance.

The problem is exacerbated when HAPPO serves as the primary model. Its extremely sharp distribution leaves virtually no room for the auxiliary model to make corrections, causing the fusion result to degenerate into a mere copy of the primary model's output. This shows that the root cause of fusion failure lies not in the fusion algorithm itself, but in the overly deterministic nature of the constituent policies, which lack the requisite diversity and complementarity. The solution is diversity-aware retraining, which encourages models to retain probability mass on suboptimal actions while maintaining high performance. After such retraining, the model distributions indeed exhibit greater diversity, creating the necessary conditions for successful fusion and leading to significant improvements (As shown in Figure 4b, 4d).

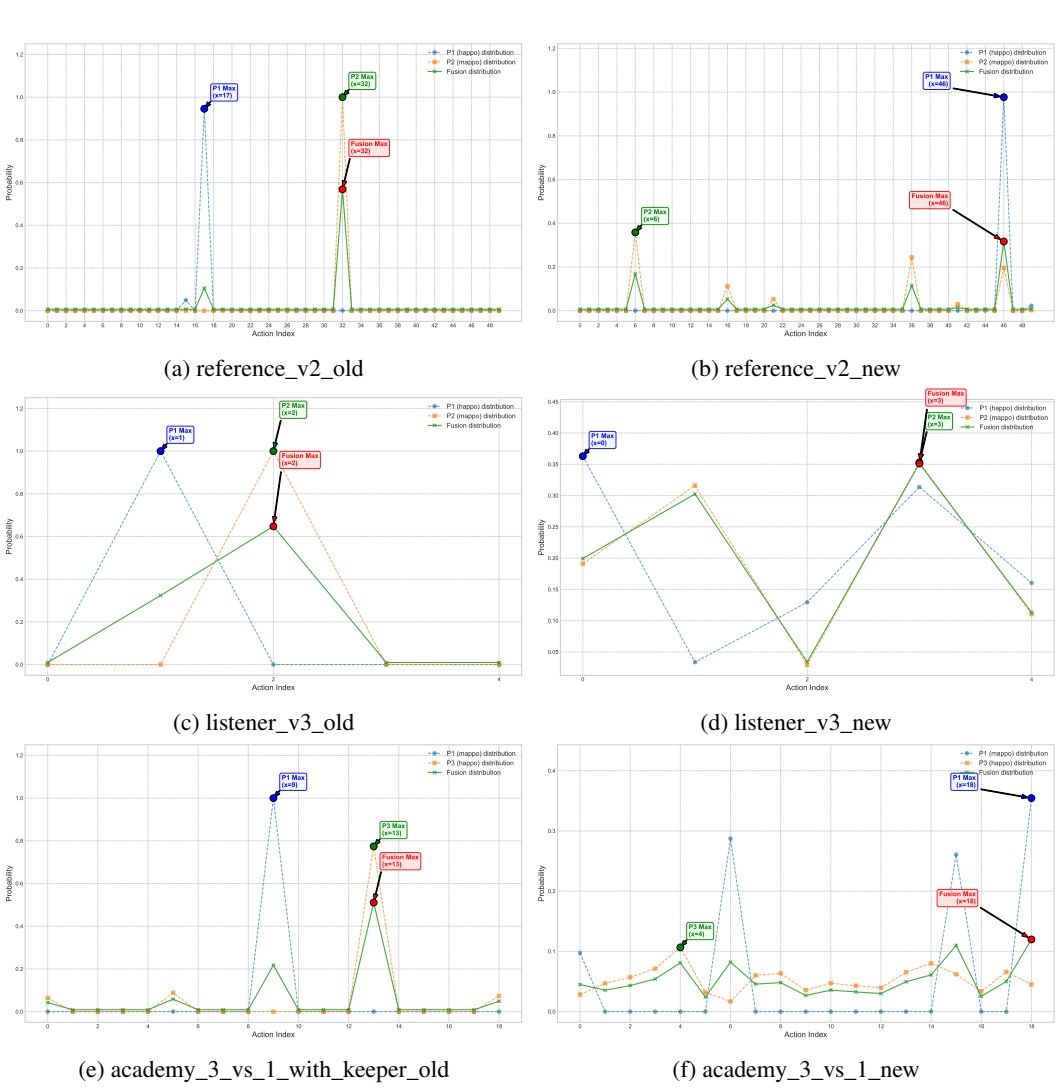

(a) reference_v2_old  (b) reference_v2_new

(c) listener_v3_old  (d) listener_v3_new

(e) academy_3_vs_1_with_keeper_old  (f) academy_3_vs_1_new

Figure 4: Visualization of Policy Distributions Before and After Co-design.This figure presents a comparison of the base models' action probability distributions before and after our co-design re-training process across three different scenarios. The left column (a, c, e) displays the original (_old) models, which exhibit highly 'peaky' or 'over-confident' distributions. Their probability mass is almost entirely concentrated on a single action, providing very limited complementary information for fusion. In contrast, the right column (b, d, f) shows the retrained (_new) models, whose distributions are markedly smoother, assigning non-trivial probabilities to multiple suboptimal actions. The increased diversity and smoothness in the _new policies create the necessary complementary information for our fusion mechanism to facilitate subsequent integration.

### A.3.2 GRF Environment

**To further investigate** the prerequisites for effective policy fusion **discussed in the main text**, we examine the Google Research Football (GRF) environment. In contrast to MPE, the policy distributions in the Google Research Football (GRF) environment are not completely extremized; suboptimal actions retain a non-trivial probability mass, suggesting the presence of latent complementarity. However, initial experiments involving direct fusion of pretrained models still resulted in failure. To validate our hypothesis of latent complementarity, we first designed two preprocessing methods as exploratory countermeasures. The first, **Top-K Confidence Alignment (Top-K Align)**, mitigates the tendency of over-confident models by reducing the peak probability of the more confident distribution to match that of the other. The second, **Top-K Intersection Filtering (Top-K Intersect)**, seeks consensus by retaining only the actions present in the intersection of their Top-K candidate sets.

While these methods led to performance gains, they are essentially symptomatic solutions. Their true value was in demonstrating that exploitable complementary information does exist within the GRF models, providing a strong justification for a more fundamental approach. We therefore adopted a **co-design** philosophy: retraining the base models to make them explicitly **"fusion-ready."** A key objective of our approach is to transform the initially peaked policy distributions, as shown in Figure 4e, into smoother, more diverse distributions, exemplified by Figure 4f.Specifically, we halted training after the models reached SOTA-level performance to prevent overfitting, encouraging them to maintain smoother probability distributions. After retraining, the models retained significantly more information in their suboptimal actions, drastically enhancing their complementarity and creating the ideal conditions for the **KPF mechanism** to deliver substantial performance gains.The methods described above **directly correspond to** the preprocessing and co-design strategies mentioned in **the main text** and **demonstrate** how we operationalize them to exploit latent complementary information in GRF.

### A.3.3 SMAC and SMACv2 Environments

KPF's outstanding performance in the SMAC and SMACv2 environments further validates the key prerequisites for effective fusion. Similar to GRF, the policy distributions in these environments are relatively smooth, providing the necessary complementary signals. Concurrently, these environments typically involve a larger number of agents and higher decision dimensionality. This increase in agent numbers not only expands the space for corrective fusion but also statistically increases the "effective corrective hit rate," making the benefits of fusion more pronounced. Furthermore, the KPF mechanism itself is inherently **stable and conservative**, operating on the principle of "correcting only when confident." It applies adjustments primarily when policy distributions are significantly different and the direction for correction is clear, while remaining conservative when uncertainty is high. This approach guarantees a performance baseline while safely exploring opportunities for improvement. Our method ensures that, even when the performance gap between models is large, the fused performance almost never degrades, though improvements may not always be guaranteed. For example(as shown in Figure7), in the **3s5z_vs_3s6z** scenario with seed = 4 (77% vs. 53%) and seed = 5 (81% vs. 60%), the models differ by more than **20%**. Yet, our KPF approach still achieves substantial improvements, boosting win rates to **88%** and **95%**, respectively. This demonstrates **the robustness of our method**: it not only guarantees a reliable lower bound on performance but also consistently leverages complementary information from weaker models to refine the stronger ones.

### A.3.4 Conclusion

Through a systematic analysis across the MPE, GRF, and SMAC/SMACv2 environments, we conclude that effective fusion depends not only on the fusion algorithm itself but more critically on the characteristics of the base models and the mechanism's design. The core prerequisites are as follows:

**Probability Distribution Smoothness.** The policy distributions must retain sufficient information in suboptimal actions to allow for the formation of complementary signals between different models.

**High Performance of Base Models.** A foundational assumption is that the individual models participating in the fusion must themselves possess a sufficiently high level of performance. If a

constituent model performs poorly, its noise or biases will be incorporated into the fused result, potentially degrading overall performance rather than enhancing it.

**A Robust Fusion Mechanism.** A robust mechanism is essential for guaranteeing a performance baseline and enabling upside potential. Robustness implies that the fusion method can maintain performance at least as good as the best individual model, even in the face of distributional shifts. Mechanisms like KPF, which use dynamic, uncertainty-aware updates, are designed to ensure this "lower-bound safety" and "upper-bound potential."

These three prerequisites are complementary, with their logic stemming from the principles of model quality control and fusion method design. Smooth distributions and high-performing models ensure the **reliability of the inputs**, while a robust fusion mechanism ensures the **resilience of the fusion strategy**. Only when these conditions are met can policy fusion realize its true potential in multi-agent reinforcement learning.

### A.4 SUPPLEMENTARY RESULTS FOR HYPERPARAMETER ABLATION STUDIES

This section provides more complete experimental results for the hyperparameter ablation study discussed in the main text. Table 7 details the specific impact of tuning key hyperparameters on KPF's performance across multiple SMAC maps and random seeds.

The data consistently validates the effectiveness of our prior-guided tuning strategy. For instance, the results for the `3s5z_vs_3s6z` (Seed 4) map clearly illustrate our staged tuning method: first, a coarse-grained search for $\alpha$ identifies a strong baseline (an 82.0% win rate at $\alpha = 0.2$); subsequently, holding $\alpha$ constant and fine-tuning the `Noise Level` pushes the performance to its peak (88.0%). Furthermore, the table highlights the high-performance ceiling of our method, as KPF is able to achieve a perfect 100% win rate on challenging maps such as `10m_vs_11m`, `27m_vs_30m`, and `8m_vs_9m` with optimal hyperparameter configurations(through slight parameter adjustments, the win rate can be further improved from 99% to 100%). From the table 7, it can be observed that variations in the parameter lead to fluctuations in win rate; however, the performance remains stable without significant degradation. Notably, the optimal result is achieved at a specific parameter value. These comprehensive results confirm that although model performance is sensitive to hyperparameters, our systematic tuning approach can reliably identify configurations that yield state-of-the-art performance in diverse multi-agent tasks.

We strictly adhered to the hyperparameter settings recommended in the original publications for all baseline methods. Nevertheless, readers may notice minor discrepancies between some of our reproduced win rates and the values reported in the original papers. This is a widely recognized phenomenon in the deep reinforcement learning community, and these differences do not stem from errors in implementation or hyperparameter selection. Instead, they are attributable to a combination of factors, primarily: (1) **Hardware Differences**: different GPU models can have slight variations in floating-point arithmetic; and (2) **Software Versions**: discrepancies in the versions of underlying libraries such as PyTorch, CUDA, and cuDNN can affect random number generation and computational precision. The results we report herein are those that have been stably reproduced on our experimental platform.

| Map | Seed | $\alpha$ | $r_2$ (Aux/Pri Ratio) | Noise Level | Win Rate (%) |
|---|---|---|---|---|---|
| 3s5z_vs_3s6z | 4 | 0.0 | 0.77/0.53 | 0.000 | 76.0 |
| 3s5z_vs_3s6z | 4 | 0.0 | 0.77/0.53 | 0.001 | 76.0 |
| 3s5z_vs_3s6z | 4 | 0.0 | 0.77/0.53 | 0.002 | 82.0 |
| 3s5z_vs_3s6z | 4 | 0.0 | 0.77/0.53 | 0.003 | 75.0 |
| 3s5z_vs_3s6z | 4 | 0.0 | 0.77/0.53 | 0.004 | **88.0** |
| 3s5z_vs_3s6z | 4 | 0.0 | 0.77/0.53 | 0.005 | 82.0 |
| 3s5z_vs_3s6z | 5 | 0.00 | 0.81/0.60 | 0.000 | 75.0 |
| 3s5z_vs_3s6z | 5 | 0.10 | 0.81/0.60 | 0.000 | 80.0 |
| 3s5z_vs_3s6z | 5 | 0.20 | 0.81/0.60 | 0.000 | 77.0 |
| 3s5z_vs_3s6z | 5 | 0.30 | 0.81/0.60 | 0.000 | 87.0 |
| 3s5z_vs_3s6z | 5 | 0.30 | 0.81/0.60 | 0.001 | **95.0** |
| 3s5z_vs_3s6z | 5 | 0.30 | 0.81/0.60 | 0.002 | 84.0 |
| 3s5z_vs_3s6z | 5 | 0.40 | 0.81/0.60 | 0.000 | 77.0 |
| MMM2 | 1 | 0.0 | 0.90/0.95 | 0.000 | **98.0** |
| corridor | 5 | 0.0 | 0.94/0.82 | 0.000 | 90.0 |
| corridor | 5 | 0.0 | 0.94/0.82 | 0.001 | 96.0 |
| corridor | 5 | 0.0 | 0.94/0.82 | 0.002 | **99.0** |
| corridor | 5 | 0.0 | 0.94/0.82 | 0.003 | 95.0 |
| corridor | 5 | 0.0 | 0.94/0.82 | 0.005 | 96.0 |
| corridor | 5 | 0.0 | 0.94/0.82 | 0.010 | 96.0 |
| 27m_vs_30m | 1 | 0.0 | 0.94/0.78 | 0.000 | 96.0 |
| 27m_vs_30m | 1 | 0.1 | 0.94/0.78 | 0.000 | 97.0 |
| 27m_vs_30m | 1 | 0.3 | 0.94/0.78 | 0.000 | 97.0 |
| 27m_vs_30m | 1 | 0.4 | 0.94/0.78 | 0.000 | **100.0** |
| 27m_vs_30m | 1 | 0.5 | 0.94/0.78 | 0.000 | 95.0 |
| 10m_vs_11m | 2 | 0.0 | 0.97/0.80 | 0.000 | 97.0 |
| 10m_vs_11m | 2 | 0.0 | 0.97/0.80 | 0.001 | 99.0 |
| 10m_vs_11m | 2 | 0.0 | 0.97/0.80 | 0.003 | 99.0 |
| 10m_vs_11m | 2 | 0.0 | 0.97/0.80 | 0.004 | 98.0 |
| 10m_vs_11m | 2 | 0.0 | 0.97/0.80 | 0.006 | 98.0 |
| 10m_vs_11m | 2 | 0.0 | 0.97/0.80 | 0.007 | 99.0 |
| 10m_vs_11m | 2 | 0.0 | 0.97/0.80 | 0.008 | **100.0** |
| 10m_vs_11m | 2 | 0.0 | 0.97/0.80 | 0.009 | 98.0 |
| 8m_vs_9m | 0 | 0.0 | 0.95/0.83 | 0.000 | 96.0 |
| 8m_vs_9m | 0 | 0.0 | 0.95/0.83 | 0.002 | 96.0 |
| 8m_vs_9m | 0 | 0.0 | 0.95/0.83 | 0.003 | 99.0 |
| 8m_vs_9m | 0 | 0.0 | 0.95/0.83 | 0.004 | 96.0 |
| 8m_vs_9m | 0 | 0.0 | 0.95/0.83 | 0.006 | **100.0** |
| 8m_vs_9m | 0 | 0.0 | 0.95/0.83 | 0.007 | 97.0 |
| 8m_vs_9m | 0 | 0.0 | 0.95/0.83 | 0.009 | 96.0 |

Table 7: Ablation study on hyperparameters across multiple SMAC maps and seeds. The $r_2$ column shows the win-rate ratio of the auxiliary policy to the primary policy.

