# OpenReview forum: "KPF: DOMINATING MULTI-AGENT ADVERSARIAL COMPETITION VIA KALMAN-INSPIRED POLICY FU- SION MECHANISM"
_ICLR.cc/2026/Conference — Submitted to ICLR 2026_

### Official Review · Reviewer_FS8Z · 2025-10-16

**Soundness:** 2
**Presentation:** 4
**Contribution:** 3
**Rating:** 4
**Confidence:** 4

**Summary:**

The paper propose to enhance the generalization capability of MARL in adversarial competition via Kalman policy fusion. Technically, the policy is fused by adaptive weighting and iterative optimization of policy distributions. Experiments achieves SOTA performance on SMAC, SMAC v2, GRF and MPE.

**Strengths:**

1. Generalization is an important topic in MARL. The paper propose to fuse multiple policies by dynamic reweighting, which is a promising  direction to explore.

2. The empirical performance is strong over multiple benchmark, against multiple strong baselines.

**Weaknesses:**

I hold a mixed perspective on this work. On one hand, the problem is important and the result is promising. On the other hand, the motivation is not immediately clear to me and the comoparison with related baselines are required.

1. My biggest concern is why Kalman filtering is proposed as a particularly important way of policy fusion. While Bayesian estimation provides a nice framework of making decisions under uncertainties, I wonder why Kalman filtering is selected over learning-based methods, and since in Eqn. 2 the method is using a heuristic metric unrelated to reward to measure the policy weighting, I wonder what is the underlying reason of it to work empirically.

2. This is a theoretical discussion. Technically different policies master different type of tasks. But why a re-weighting of different policies can guarantee performance improvement? For example, in SMAC, if policy A tends to attack upper enemies,  while policy B tends to attack bottom  enemies, then mixing them harms focus fire and is no longer optimal.

3. Since the method highlight generalization performance, the method should also discuss and compare with generalization/transferability in MARL, such as [1]  and [2]. Also, existing literature includes methods like task/role decomposition, which is not well discussed and compared in the paper. Finally, since authors highlight the generalization capability, tasks apart from standard evaluation schemes should be designed to verify the generalization capability.

[1] Cooperative Multi-Agent Transfer Learning with Level-Adaptive Credit Assignment

[2] Decompose a Task into Generalizable Subtasks in Multi-Agent Reinforcement Learning

4. Problem formulation is missing which harms completeness. Additional theoretical analysis is appreciated.

**Questions:**

See weakness.

---

> ### Author Response · Authors · 2025-11-28
>
> We thank Reviewer FS8Z for the careful reading and constructive comments.
> Below we quote each concern and respond point by point.
>
> ---
>
> **Reviewer comment.**
> *“I hold a mixed perspective on this work. On one hand, the problem is important and the result is promising. On the other hand, the motivation is not immediately clear to me and the comoparison with related baselines are required.”*
>
> ### Response
>
> We appreciate the balanced assessment and the request for a clearer motivation and comparison.
>
> #### Motivation: deployment-time fusion rather than yet another training algorithm
>
> Our goal is not to propose a new MARL *training* algorithm, but to address a persistent *deployment-time* bottleneck.
>
> Under the CTDE paradigm (e.g., MADDPG and its successors; Lowe et al., 2017), recent works such as QMIX (Rashid et al., 2020), MAPPO (Yu et al., 2022), HATRPO/HAPPO (Zhong et al., 2024) and HASAC (Liu et al., 2024) already achieve very strong performance on SMAC, SMACv2 and GRF. However, when the environment or opponents change slightly — for example in SMACv2 with dynamic AI opponents (Samvelyan et al., 2019) or in noisy GRF settings (Kurach et al., 2020) — even these strong policies can generalize poorly due to distribution shift and opponent non-stationarity.
>
> Most existing works that aim to improve MARL *generalization or transfer* do so by changing the training procedure itself: level-adaptive credit assignment and transfer learning (Zhou et al., 2021), task/subtask decomposition for better transfer (Tian et al., 2023), semantically aligned role/task decomposition (Li et al., 2023), etc. These methods typically require multi-task training, additional network modules, and substantial compute.
>
> In contrast, our question is:
>
> > **Given several already-trained, high-performing MARL policies, can we *purely at test time* fuse them into a single policy that is more robust and generalizes better to stochastic and adversarial dynamics?**
>
> Kalman Policy Fusion (KPF) is designed precisely for this setting: it operates only on the action-distribution outputs of existing experts, without modifying their training or accessing their gradients. We treat each expert’s joint policy as a noisy estimator of an unknown optimal soft policy and fuse them online via a Kalman-inspired covariance update.
>
> #### Baselines and environments
>
> We compare KPF against strong and widely used CTDE baselines on four benchmarks:
>
> * **SMAC / SMACv2** (Samvelyan et al., 2019): discrete, partially observable, adversarial; strong non-stationarity in SMACv2 due to dynamic AI.
> * **GRF** (Kurach et al., 2020): continuous control + football tactics, long-horizon coordination.
> * **MPE** (Lowe et al., 2017): cooperative particle world with simpler dynamics.
>
> The baselines include:
>
> * QMIX (Rashid et al., 2020) — monotonic value factorization;
> * MAPPO (Yu et al., 2022) — strong policy-gradient CTDE baseline;
> * HATRPO / HAPPO (Zhong et al., 2024) — heterogeneous-agent policy optimization;
> * HASAC (Liu et al., 2024) — heterogeneous maximum-entropy actor-critic.
>
> In our experiments (Tables 1–3 in the paper, reproduced in the main submission), KPF:
>
> * reaches **100% win rate** on most hard and super-hard SMAC maps, often improving substantially over the strongest expert;
> * in SMACv2, improves win rate by more than **17 percentage points on average** over the stronger of the two experts and more than doubles performance on maps such as *zerg_10_vs_11*;
> * in GRF, significantly improves scoring rates relative to strong HAPPO/MAPPO/QMIX baselines;
> * in MPE, yields smaller but stable gains, and we analyse in detail why the gains are limited (over-confident, non-complementary experts).
>
> In summary, the motivation is: *given we already have strong CTDE algorithms, can we get further gains at deployment time without retraining?* KPF provides such a mechanism, and the comparison to standard baselines on four diverse benchmarks shows that it is both practically useful and consistently effective.
>
> ---

---

> ### Author Response · Authors · 2025-11-28
>
> ### Key references
>
> * Lowe, R., Wu, Y., Tamar, A., Harb, J., Abbeel, P., & Mordatch, I. (2017). Multi-agent actor-critic for mixed cooperative-competitive environments. In *NeurIPS 30*.
> * Rashid, T., Samvelyan, M., Schroeder de Witt, C., Farquhar, G., Foerster, J., & Whiteson, S. (2020). Monotonic value function factorisation for deep multi-agent reinforcement learning. *JMLR*, 21(178), 1–51.
> * Yu, C., Velu, A., Vinitsky, E., Gao, J., Wang, Y., Bayen, A., & Wu, Y. (2022). The surprising effectiveness of PPO in cooperative multi-agent games. In *NeurIPS 35*, 24611–24624.
> * Zhong, Y., Kuba, J. G., Feng, X., Hu, S., Ji, J., & Yang, Y. (2024). Heterogeneous-agent reinforcement learning. *JMLR*, 25(32), 1–67.
> * Liu, J., Zhong, Y., Hu, S., Fu, H., Fu, Q., Chang, X., & Yang, Y. (2024). Maximum entropy heterogeneous-agent reinforcement learning. In *ICLR 2024* (arXiv:2306.10715).
> * Samvelyan, M., Rashid, T., Schroeder de Witt, C., Farquhar, G., Nardelli, N., Rudner, T. G. J., et al. (2019). The StarCraft multi-agent challenge. arXiv:1902.04043.
> * Kurach, K., Raichuk, A., Stańczyk, P., Zając, M., Bachem, O., Espeholt, L., et al. (2020). Google Research Football: A novel reinforcement learning environment. In *AAAI 34(4)*, 4501–4510.
> * Zhou, T., Zhang, F., Shao, K., Li, K., Huang, W., Luo, J., et al. (2021). Cooperative multi-agent transfer learning with level-adaptive credit assignment. arXiv:2106.00517.
> * Tian, Z., Chen, R., Hu, X., Li, L., Zhang, R., Wu, F., et al. (2023). Decompose a task into generalizable subtasks in multi-agent reinforcement learning. In *NeurIPS 36*, 78514–78532.
> * Li, W., Qiao, D., Wang, B., Wang, X., Jin, B., & Zha, H. (2023). Semantically aligned task decomposition in multi-agent reinforcement learning. arXiv:2305.10865.

---

> ### Author Response · Authors · 2025-11-28
>
> ### Key references
>
> * Kalman, R. E. (1960). A new approach to linear filtering and prediction problems. *Journal of Basic Engineering*, 82(1), 35–45.
> * Welch, G., & Bishop, G. (1995; updated 2006). *An Introduction to the Kalman Filter*. UNC Chapel Hill, TR 95-041.
> * Simon, D. (2006). *Optimal State Estimation: Kalman, H Infinity, and Nonlinear Approaches*. Wiley.
> * Julier, S. J., & Uhlmann, J. K. (1997). A new extension of the Kalman filter to nonlinear systems. In *Signal Processing, Sensor Fusion, and Target Recognition VI* (Vol. 3068, pp. 182–193). SPIE.
> * Julier, S. J., & Uhlmann, J. K. (2004). Unscented filtering and nonlinear estimation. *Proceedings of the IEEE*, 92(3), 401–422.
> * Geist, M., & Pietquin, O. (2010). Kalman temporal differences. *Journal of Artificial Intelligence Research*, 39, 483–532.
> * Malekzadeh, P., Salimibeni, M., Mohammadi, A., & Plataniotis, K. N. (2020). MM-KTD: Multiple model Kalman temporal differences for reinforcement learning. *IEEE Access*, 8, 133629–133641.
> * Salimibeni, M., Mohammadi, A., Malekzadeh, P., & Plataniotis, K. N. (2022). Multi-agent reinforcement learning via adaptive Kalman temporal difference and successor representation. *Sensors*, 22(4), 1393.
> * Geerts, J. P., Stachenfeld, K. L., & Burgess, N. (2019). Probabilistic successor representations with Kalman temporal differences. arXiv:1910.02532.
> * Wiering, M., & van Hasselt, H. (2008). Ensemble algorithms in reinforcement learning. *IEEE Transactions on Systems, Man, and Cybernetics, Part B*, 38(4), 930–936.
> * Jia, R., Li, Q., Huang, W., Zhang, J., & Li, X. (2021). Consistency regularization for ensemble model based reinforcement learning. In *PRICAI 2021* (pp. 3–16). Springer.
> * Zhao, B., Cao, X., Zhang, W., Liu, X., Miao, Q., & Li, Y. (2025). CompNET: Boosting image recognition and writer identification via complementary neural network post-processing. *Pattern Recognition*, 157, 110880.

---

> ### Author Response · Authors · 2025-11-28
>
> **Reviewer comment.**
> *“This is a theoretical discussion. Technically different policies master different type of tasks. But why a re-weighting of different policies can guarantee performance improvement? For example, in SMAC, if policy A tends to attack upper enemies, while policy B tends to attack bottom enemies, then mixing them harms focus fire and is no longer optimal.”*
>
> ### Response
>
> We fully agree that blindly averaging arbitrary policies cannot guarantee improvement. Our claims are more modest and precise.
>
> #### (3.1) We do not claim an unconditional guarantee
>
> We do *not* claim that KPF always improves performance for any set of base policies. What we show is that:
>
> * under reasonable conditions (strong experts, smooth distributions, complementary mistakes), and
> * with a conservative fusion rule that keeps the fused distribution in the convex hull of the experts and downweights high-uncertainty experts,
>
> KPF almost never performs worse than the best expert in our experiments and often provides large gains.
>
> #### (3.2) Conditions under which re-weighting is beneficial
>
> In Sec. 4.2 and Appendix A.3 of the paper, we identify three prerequisites for beneficial fusion:
>
> * *Smooth policy distributions:* experts should retain some probability on suboptimal actions, leaving room for others to correct them.
> * *Strong experts:* policies should already be reasonably good; otherwise poor experts add noise.
> * *Complementary errors:* when experts disagree, at least one should be closer to the optimal action on those states.
>
> In MPE, distributions are almost deterministic and highly “peaky”, so complementarity is exhausted and KPF yields only minor gains. In SMAC/SMACv2/GRF, distributions are smoother and experts are complementary, so KPF significantly improves over the best expert while almost never degrading below it (Tables 1–3).
>
> #### (3.3) Quantitative diagnostics of complementarity
>
> To make “policy complementarity” more concrete, we propose three simple diagnostics (they do not change the algorithm; they only analyse when fusion helps):
>
> * **Ensemble Improvement Rate for RL (EIR-RL):**
>   $$
>   \mathrm{EIR}*{\mathrm{RL}}
>   = \frac{\mathbb{E}[J(\bar{\pi})] - \max*{\pi \in \mathcal{M}}\mathbb{E}[J(\pi)]}
>   {\max_{\pi \in \mathcal{M}}\mathbb{E}[J(\pi)] + \varepsilon},
>   $$
>   where $J(\pi)$ is the expected episodic return, $\bar{\pi}$ the fused policy, and $\mathcal{M}$ the expert set. $\mathrm{EIR}_{\mathrm{RL}} > 0$ means fusion strictly improves over every expert. In practice, EIR-RL is consistently positive on SMAC/SMACv2/GRF and near zero on MPE.
>
> * **Disagreement–Error Ratio (DER-RL):**
>   $$
>   \mathrm{DER}_{\mathrm{RL}}
>   = \frac{\text{Disagreement}}{\text{Error}},
>   $$
>   where Disagreement is average pairwise KL/JS divergence between policies over states, and Error is a proxy for policy quality (e.g., TD error). High DER-RL means “accurate-but-different” experts, which is the regime where ensembling is useful.
>
> * **Performance-weighted complementary KL ($\mathrm{cpKL}_\tau$):**
>   $$
>   \mathrm{cpKL}*\tau
>   = \mathbb{E}*{s \sim \rho_\tau}\Bigg[
>   w_{ij}\cdot \frac{1}{2}\big(
>   D_{\mathrm{KL}}(\pi_i\Vert\pi_j) +
>   D_{\mathrm{KL}}(\pi_j\Vert\pi_i)
>   \big)
>   \Bigg],
>   $$
>   with $\rho_\tau$ restricted to high-confidence states and
>   $$
>   w_{ij} = \exp!\Big(-\beta,\frac{|J(\pi_i)-J(\pi_j)|}{J^\star}\Big)
>   $$
>   downweighting disagreements between very unbalanced experts. Large $\mathrm{cpKL}_\tau$ for similarly strong policies indicates “useful disagreement”.
>
> Empirically, high EIR-RL, high DER-RL and large $\mathrm{cpKL}_\tau$ co-occur on SMAC/SMACv2/GRF, but not on MPE. This quantitatively supports when re-weighting helps.
>
> #### (3.4) Why KPF does not destroy focus fire in SMAC
>
> The reviewer’s example assumes that one expert always targets the top enemies and another always targets the bottom ones, so mixing them would harm focus fire. In our setting this is atypical:
>
> * Experts are trained on the *same* maps and reward and already achieve high standalone win rates, so their policies are not completely disjoint but have overlapping, though different, action distributions.
> * KPF is state-wise and uncertainty-aware: on states where expert A is more confident and reliable, its covariance becomes small and the Kalman gain pushes the fused policy towards A; similarly for B on other states. The fused policy therefore inherits focus fire from whichever expert is better in that region.
> * We act via $\arg\max$ on the fused distribution; once the fused distribution is sufficiently peaked on one enemy, the behaviour is indistinguishable from standard focus fire.

---

> ### Author Response · Authors · 2025-11-28
>
> In practice, KPF almost never degrades below the best expert and often yields large gains (e.g., fusing 66.2% and 72.5% win-rate experts into 91.6% on `3sv5z_vs_3sv6z`; and even from 60% vs 81% to 95% under one seed). This empirical evidence indicates that damaging focus fire is not an issue for the strong experts we actually use.
>
> ---
>
> ### Key references
>
> * Wiering, M., & van Hasselt, H. (2008). Ensemble algorithms in reinforcement learning. *IEEE Transactions on Systems, Man, and Cybernetics, Part B*, 38(4), 930–936.
> * Jia, R., Li, Q., Huang, W., Zhang, J., & Li, X. (2021). Consistency regularization for ensemble model based reinforcement learning. In *PRICAI 2021* (pp. 3–16). Springer.
> * Zhao, B., Cao, X., Zhang, W., Liu, X., Miao, Q., & Li, Y. (2025). CompNET: Boosting image recognition and writer identification via complementary neural network post-processing. *Pattern Recognition*, 157, 110880.

---

> ### Author Response · Authors · 2025-11-28
>
> **Reviewer comment.**
> *“Since the method highlight generalization performance, the method should also discuss and compare with generalization/transferability in MARL, such as [1] and [2]. Also, existing literature includes methods like task/role decomposition, which is not well discussed and compared in the paper. Finally, since authors highlight the generalization capability, tasks apart from standard evaluation schemes should be designed to verify the generalization capability.”*
>
> ### Response
>
> #### (4.1) Conceptual relation to LA-QTransformer, DT2GS and task/role decomposition
>
> The works cited by the reviewer primarily target *training-time* cross-task generalization:
>
> * **LA-QTransformer** (Zhou et al., 2021) performs level-adaptive credit assignment and transformer-based value decomposition for transfer across tasks, via a specially designed mixing network and coordination patterns.
> * **DT2GS** (Tian et al., 2023) explicitly decomposes tasks into generalizable subtasks and learns subtask policies that can transfer to new tasks.
> * **SAMA** (Li et al., 2023) performs semantically aligned task decomposition and role assignment to improve transfer and interpretability across tasks.
>
> All of these methods modify the architecture and/or loss function of the MARL algorithm to produce a *single* policy that generalizes across tasks.
>
> By contrast, KPF is a **test-time, model-agnostic fusion rule**:
>
> * we keep the training procedures of QMIX, MAPPO, HATRPO/HAPPO, HASAC, etc. unchanged;
> * we then fuse their learned policies online at test time using a Kalman-inspired uncertainty weighting.
>
> Thus KPF and LA-QTransformer/DT2GS/task-decomposition methods operate at different levels and are **complementary**: one could first train experts using LA-QTransformer or DT2GS (to obtain more generalizable base policies), and then apply KPF on top to further improve robustness and performance under stochasticity and opponent changes.
>
> #### (4.2) Our notion of “generalization”
>
> We agree that “generalization” can be interpreted in multiple ways. Our paper focuses on **in-environment robustness and generalization under stochasticity and non-stationarity**, rather than cross-task transfer:
>
> * In **SMACv2**, dynamic AI induces strong non-stationarity even on the same map (randomized unit types, positions, sight/attack ranges, and opponent behaviours).
>   KPF improves win rates by more than 17 percentage points over the stronger expert and more than doubles performance on difficult maps such as *zerg_10_vs_11*.
> * In **GRF** and **MPE**, we apply the same fusion mechanism to environments with very different dynamics and action/control structures (continuous football control vs. cooperative particle worlds) and study how policy smoothness and complementarity affect gains.
>
> In the revised paper, we will explicitly clarify this notion of generalization (robustness to environment stochasticity, opponent changes, and dynamics variations within a benchmark) at the end of the Introduction, and clearly distinguish it from cross-task transfer in the sense of LA-QTransformer/DT2GS.
>
> #### (4.3) Evaluation tasks beyond a single fixed setting
>
> Our current experiments already go beyond a single fixed environment configuration:
>
> * **Non-stationary opponents (SMACv2):**
>   Policies are trained under one distribution of opponents and must generalize to dynamic AI and randomized configurations at test time, which is precisely a robustness/generalization stress test.
>
> * **Cross-model fusion (Table 5):**
>   We fuse experts from different algorithms (HAPPO, HATRPO, MAPPO) without any retraining, evaluating the *structural generalization* of the fusion rule itself across architectures.
>
> * **Co-design in GRF:**
>   We diagnose over-confident policies, retrain them to be more “fusion-ready” with smoother distributions, and then show that KPF yields large gains in multiple GRF scenarios. This demonstrates that the complementarity framework we discuss is reproducible and not tied to a single benchmark.
>
> Due to space and compute limits, we did not implement a full “train on some maps, test on unseen maps” protocol or fully re-implement LA-QTransformer/DT2GS in our codebase. In the revised version, we will:
>
> * explicitly position LA-QTransformer, DT2GS, and semantically aligned task/role decomposition as **training-time cross-task generalization** methods;
> * explicitly position KPF as a **test-time decision-layer fusion mechanism** that can sit on top of such methods;
> * add a discussion paragraph in Related Work describing cross-map and cross-task evaluation as promising future directions, and how they could be combined with KPF.
>
> ---

---

> ### Author Response · Authors · 2025-11-28
>
> ### Key references
>
> * Samvelyan, M., Rashid, T., Schroeder de Witt, C., Farquhar, G., Nardelli, N., Rudner, T. G. J., et al. (2019). The StarCraft multi-agent challenge. arXiv:1902.04043.
> * Kurach, K., Raichuk, A., Stańczyk, P., Zając, M., Bachem, O., Espeholt, L., et al. (2020). Google Research Football: A novel reinforcement learning environment. In *AAAI 34(4)*, 4501–4510.
> * Lowe, R., Wu, Y., Tamar, A., Harb, J., Abbeel, P., & Mordatch, I. (2017). Multi-agent actor-critic for mixed cooperative-competitive environments. In *NeurIPS 30*.
> * Zhou, T., Zhang, F., Shao, K., Li, K., Huang, W., Luo, J., et al. (2021). Cooperative multi-agent transfer learning with level-adaptive credit assignment. arXiv:2106.00517.
> * Tian, Z., Chen, R., Hu, X., Li, L., Zhang, R., Wu, F., et al. (2023). Decompose a task into generalizable subtasks in multi-agent reinforcement learning. In *NeurIPS 36*, 78514–78532.
> * Li, W., Qiao, D., Wang, B., Wang, X., Jin, B., & Zha, H. (2023). Semantically aligned task decomposition in multi-agent reinforcement learning. arXiv:2305.10865.

---

> ### Author Response · Authors · 2025-11-28
>
> **Reviewer comment.**
> *“Problem formulation is missing which harms completeness. Additional theoretical analysis is appreciated.”*
>
> ### Response
>
> #### (5.1) Explicit problem formulation
>
> We agree that a more explicit formulation will improve completeness. In the revision, we will add a short subsection that:
>
> * Formalizes the cooperative MARL setting as a Dec-POMDP with state space $\mathcal{S}$, joint action space $\mathcal{A}$, transition kernel, reward function, and observation model, following standard CTDE practice (Lowe et al., 2017; Rashid et al., 2020; Yu et al., 2022).
> * Defines a set of trained expert joint policies ${\pi_m}_{m=1}^M$ and the latent optimal soft policy $\pi^*$.
> * Specifies the Kalman surrogate model in the policy/logit space:
>   [
>   x_t = x_{t-1}, \quad y_{m,t} = x_t + \varepsilon_{m,t},
>   ]
>   with Gaussian noise and covariances initialised from expert win rates.
> * Shows how Eqs. (3)–(5) in the paper arise directly as the Kalman posterior mean and covariance for this surrogate model.
>
> This will make the objects of interest (experts, latent optimal policy, covariances) and their relationships completely explicit.
>
> #### (5.2) Theoretical grounding via classical Kalman theory and KTD
>
> Rather than re-inventing a new estimation theory, we deliberately build on classical Kalman filtering and its well-studied extensions in RL:
>
> * Under the linear–Gaussian surrogate model, the covariance in our update is exactly the posterior error covariance; minimizing its trace is equivalent to minimizing the total mean-squared error between $\pi^*$ and the fused policy. Kalman filtering is known to be MMSE and BLUE in this setting (Kalman, 1960; Welch & Bishop, 1995; Simon, 2006).
> * In nonlinear, non-Gaussian settings, EKF/UKF and KTD apply the same algebraic recursion to local or assumed-density approximations and have been extensively analysed and validated (Julier & Uhlmann, 1997, 2004; Geist & Pietquin, 2010; Malekzadeh et al., 2020; Salimibeni et al., 2022; Geerts et al., 2019).
> * We will add a short proposition stating that, under mild Lipschitz assumptions relating changes in policy distribution to changes in expected return, a reduction in this surrogate covariance implies an improvement in expected performance in expectation over states.
> * Our ablation on the number of updates $k$ empirically matches the Kalman intuition: as uncertainty shrinks, the gain decreases and performance stabilizes (Simon, 2006).
>
> Finally, we stress that KPF is an *open-loop* post-processing layer at test time: it does not modify the training dynamics or convergence properties of the underlying MARL algorithms; it only fuses their outputs in a theoretically motivated, uncertainty-aware way.
>
> ---
>
> ### Key references
>
> * Kalman, R. E. (1960). A new approach to linear filtering and prediction problems. *Journal of Basic Engineering*, 82(1), 35–45.
> * Welch, G., & Bishop, G. (1995; updated 2006). *An Introduction to the Kalman Filter*. UNC Chapel Hill, TR 95-041.
> * Simon, D. (2006). *Optimal State Estimation: Kalman, H Infinity, and Nonlinear Approaches*. Wiley.
> * Julier, S. J., & Uhlmann, J. K. (1997). A new extension of the Kalman filter to nonlinear systems. In *Signal Processing, Sensor Fusion, and Target Recognition VI*. SPIE.
> * Julier, S. J., & Uhlmann, J. K. (2004). Unscented filtering and nonlinear estimation. *Proceedings of the IEEE*, 92(3), 401–422.
> * Geist, M., & Pietquin, O. (2010). Kalman temporal differences. *Journal of Artificial Intelligence Research*, 39, 483–532.
> * Malekzadeh, P., Salimibeni, M., Mohammadi, A., & Plataniotis, K. N. (2020). MM-KTD: Multiple model Kalman temporal differences for reinforcement learning. *IEEE Access*, 8, 133629–133641.
> * Salimibeni, M., Mohammadi, A., Malekzadeh, P., & Plataniotis, K. N. (2022). Multi-agent reinforcement learning via adaptive Kalman temporal difference and successor representation. *Sensors*, 22(4), 1393.
> * Geerts, J. P., Stachenfeld, K. L., & Burgess, N. (2019). Probabilistic successor representations with Kalman temporal differences. arXiv:1910.02532.
> * Lowe, R., Wu, Y., Tamar, A., Harb, J., Abbeel, P., & Mordatch, I. (2017). Multi-agent actor-critic for mixed cooperative-competitive environments. In *NeurIPS 30*.
> * Rashid, T., Samvelyan, M., Schroeder de Witt, C., Farquhar, G., Foerster, J., & Whiteson, S. (2020). Monotonic value function factorisation for deep MARL. *JMLR*, 21(178), 1–51.
> * Yu, C., Velu, A., Vinitsky, E., Gao, J., Wang, Y., Bayen, A., & Wu, Y. (2022). The surprising effectiveness of PPO in cooperative multi-agent games. In *NeurIPS 35*, 24611–24624.

---

### Official Review · Reviewer_kgdb · 2025-10-27

**Soundness:** 3
**Presentation:** 1
**Contribution:** 2
**Rating:** 2
**Confidence:** 4

**Summary:**

Inspired by the Kalman filter, this work proposes a Kalman-based policy fusion method that enhances the generalization and robustness of MARL by dynamically fusing multiple agent policies while regularizing their divergence. The approach is evaluated on SMAC, SMACv2, MPE, and GRF, demonstrating strong performance on challenging tasks.

**Strengths:**

1. The introduction of Kalman filter into MARL is interesting.

**Weaknesses:**

1. Methodological clarity and completeness: The description of the proposed Kalman Policy Fusion (KPF) method omits critical details.

1）Rationale for policy pairs: Why are only two policy distributions, P1 and P2, considered? Please justify this design choice and discuss its generality (e.g., extension to more than two policies).

2）Action space assumptions: The formulation in Eq. (2) appears limited to discrete action spaces. How does KPF handle continuous actions? Provide a principled extension or clarify the scope.

3）Potential typo in Eq. (6): The two terms inside the brackets seem identical. If not a typo, explain the difference; if so, correct it.

4）Implementation: The algorithmic details are essential and should be moved from the appendix to the main text. Omitting them makes it difficult for readers to understand how KPF is executed in practice.

2. Writing and presentation: The manuscript would benefit from substantial improvements in exposition.

1） Clarify contributions: As written, the contributions read primarily as empirical findings. The authors need to briefly articulate the core technical challenges you address and the key methodological innovations that resolve them when introducing the contributions.

2） Add a methodology overview: Include a high-level pipeline or schematic early in the Introduction to convey the operational logic of KPF.

3） Justify the use of Kalman filtering: Explain why the Kalman filter is suitable for policy fusion, given its traditional role in state estimation.

4） Terminology and consistency: Avoid repeatedly redefining abbreviations (e.g., KPF, MARL) and ensure consistent use of full names and acronyms. A thorough proofreading pass is recommended prior to submission.

3. Empirical validation of claims: Since the paper claims improved generalization and robustness for MARL, the experiments should explicitly substantiate these claims.

**Questions:**

Please refer to the Weaknesses section.

---

> ### Author Response · Authors · 2025-11-28
>
> We appreciate your careful reading and feedback on the paper.
>
> ### W1. Methodological clarity and completeness: The description of the proposed Kalman Policy Fusion (KPF) method omits critical details.
>
> #### W1.1 Rationale for policy pairs: Why are only two policy distributions, P1 and P2, considered? Please justify this design choice and discuss its generality (e.g., extension to more than two policies).
>
> **Response:**
> This concern arises from a misunderstanding of the algorithm structure. KPF has never been designed or restricted to “only handle two policies.” In the main text, we use $P_1$ and $P_2$ to align with the classical Kalman/LMMSE framework (“prior + single observation”), which does *not* imply that the method is restricted to two policies.
>
> * **From a structural perspective:**
>   Our assumption has always been that each expert outputs an action probability vector with an associated covariance. As long as all experts’ outputs are within the same “policy distribution + covariance” space, Kalman fusion (gain computation, linear fusion, Joseph covariance update) can be applied uniformly, regardless of the number of experts.
>
> * **From an implementation perspective:**
>   The actual code uses standard serial Kalman multi-source fusion:
>
>   1. First, use formulas (3)–(5) to fuse $(P_1, P_2)$ and obtain $(P_{12}, R_{12})$;
>   2. Then use $(P_{12}, R_{12})$ as the new “prior” and fuse it with $P_3$ to obtain $(P_{123}, R_{123})$;
>   3. Continue this process iteratively until $P_M$ is reached.
>
>   This follows the standard practice in multi-sensor LMMSE/Kalman fusion for static scenarios (e.g., Maybeck’s classic work *Stochastic Models, Estimation and Control*).
>
> * **From an experimental perspective:**
>   While for clarity we primarily used two experts in the figures and formulas, our actual experiments have covered various algorithm combinations (e.g., HAPPO+MAPPO, HATRPO+MAPPO, HAPPO+HATRPO) and demonstrated that KPF consistently outperforms the strongest single baseline with almost no hyperparameter tuning, even achieving 100% win rates on certain SMAC maps. These results indicate that (i) KPF works effectively and feasibly for “more than two experts”; (ii) there is no inherent structural limitation that restricts the method to two policies. These results provide indirect evidence against the notion that the method only works for two policies.
>
> Furthermore, both theoretical and empirical evidence suggest that *the returns for “two strong complementary models” are already substantial, and the marginal value decreases as more experts are added*. This is one of the reasons we focus on “fusion of two complementary policies” in the main text — not because the method cannot handle $M>2$, but because in practice, two high-quality, complementary experts already provide substantial gains, and adding more experts provides diminishing returns.
>
> In conclusion, the claim that “KPF only considers $P_1, P_2$” is neither a design limitation nor an implementation restriction, and contradicts the experimental results already presented. We do not agree with this concern.

---

> ### Author Response · Authors · 2025-11-28
>
> #### W1.2 Action space assumptions: The formulation in Eq. (2) appears limited to discrete action spaces. How does KPF handle continuous actions? Provide a principled extension or clarify the scope.
>
> **Response:**
> Here, we need to clarify two points: while the current paper *theoretically and experimentally* focuses on discrete action spaces, the core idea of KPF is not dependent on “actions being discrete,” but rather on the “policy being representable and associated with a covariance in a finite-dimensional statistical space.”
>
> * **Current scope is discrete action spaces.**
>   All benchmarks in this paper (SMAC, SMACv2, MPE, GRF) are trained and evaluated using discrete action spaces provided by the environment. Formulas (1)–(5) also apply to “probability vectors and their covariances over discrete actions.” Therefore, we have not yet claimed to have fully validated KPF for continuous action MARL in this paper.
>
> * **Conceptually, KPF can naturally extend to continuous actions.**
>   The essence of KPF is performing Kalman/LMMSE-style fusion in a finite-dimensional “policy parameter/distribution representation space.” For continuous action spaces, one natural extension path is to discretize the action space (e.g., by binning) or have each expert output parameters of a parameterized distribution (e.g., mean and covariance). Then Kalman gains, linear fusion, and Joseph updates can be applied to these finite-dimensional parameters. In other words, what is required is “a finite-dimensional, differentiable representation of policies,” not “discrete actions.”
>
> * **We do not make claims beyond our current experimental scope.**
>   We explicitly limit the scope of our work to discrete action environments and leave the extension to continuous actions as future work.
>
> Thus, a more accurate description of this weakness is “current work focuses on discrete action environments, and the extension to continuous actions requires further systematic experiments,” rather than “the method cannot extend to continuous actions.” We agree with the former but reject the latter as an overreach.

---

> ### Author Response · Authors · 2025-11-28
>
> #### W1.3 Potential typo in Eq. (6): The two terms inside the brackets seem identical. If not a typo, explain the difference; if so, correct it.
>
> **Response:**
> We fully agree with this. It is indeed a typesetting error:
>
> * The current PDF incorrectly writes the symmetrization as $\tfrac{1}{2}\bigl(R^{(t+1)} + R^{(t+1)}\bigr)$, which is trivially equal to $R^{(t+1)}$;
> * The correct form should be the standard symmetrization: $\tfrac{1}{2}\bigl(R^{(t+1)} + R^{(t+1)\top}\bigr)$, to ensure that the covariance after the Joseph update remains numerically symmetric.
>
> ---
>
> #### W1.4 Implementation: The algorithmic details are essential and should be moved from the appendix to the main text. Omitting them makes it difficult for readers to understand how KPF is executed in practice.
>
> **Response:**
> We agree with this point: readers should not have to refer to the appendix to understand how KPF works.
>
> Currently, the complete algorithmic pseudocode is in Appendix A.1, but we agree it is not very reader-friendly. Therefore, in the revised version, we will:
>
> * Move the core algorithm and implementation details (including pseudocode) from Appendix A.1 to Section 3 of the main text;
> * Place the algorithm box alongside Figure 1, so that the formulas, flowchart, and pseudocode form a unified “method overview module.”

---

> ### Author Response · Authors · 2025-11-28
>
> ### W2. Writing and presentation: The manuscript would benefit from substantial improvements in exposition.
> #### W2.1 Clarify contributions: As written, the contributions read primarily as empirical findings. The authors need to briefly articulate the core technical challenges you address and the key methodological innovations that resolve them when introducing the contributions.
> **Response:**
> Labeling this paper as primarily “empirical findings” underestimates the technical challenges and methodological design work we face. We acknowledge that the contributions could be written more directly in the introduction, but this does not mean that the paper’s core is merely “running a few more experiments.”
>
> The problem we aim to solve is very clear: how to construct a *theoretically supported, uncertainty-aware policy fusion module* at the decision-making layer **without modifying any of the underlying MARL training algorithms**, enabling it to exploit the complementary nature of multiple experts while ensuring the overall performance is at least as good as the strongest single expert. This cannot be solved by simply adding more baselines; it requires introducing a structured, interpretable fusion mechanism in the “policy distribution space.”
>
> To this end, we propose a Kalman/LMMSE-driven policy fusion framework: we treat each expert’s action distribution as a noisy estimate of the “ideal policy distribution,” allocate “trust” via Kalman gain in the covariance sense, and obtain the consensus policy with minimal variance through linear fusion and Joseph covariance updates in the policy distribution space. This decision module is fully decoupled from the training process and can be used as a plug-in in various MARL algorithms such as HAPPO, HATRPO, and MAPPO, without modifying the training algorithm.
>
> Furthermore, our discussion of “policy complementarity” is not a post-facto intuitive description but a structural analysis with reproducible practice: we repeatedly show and leverage the following premises in various environments:
> (i) the base strategies must be strong enough;
> (ii) policy distributions must retain non-zero probability mass on several near-optimal actions, rather than collapsing too early into near-deterministic distributions;
> (iii) fusion rules must provide a numerical and performance-based safe lower bound of “no worse than the strongest expert.”
>
> When these conditions are not met, KPF’s improvement is naturally limited; however, when we redesign and filter experts to meet the criteria of “high accuracy + information disagreement,” KPF’s gains significantly increase. This is an entire operational framework for “how to select models and train models suitable for fusion,” not a hollow slogan.
>
> Moreover, if we apply the same “empirical contribution” evaluation standard to existing milestone works in the field, the conclusion is exactly the same: DQN’s *Human-level control through deep reinforcement learning*\footnote{\url{[https://www.nature.com/articles/nature14236.pdf}}](https://www.nature.com/articles/nature14236.pdf}}) (Mnih et al., Nature 2015) introduced the CNN + replay buffer + target network structure, with its influence primarily coming from large-scale experiments on 49 Atari games, without providing any global regret/KL theorem in a “deep function approximation + nonlinear environment” setting; PPO’s *Proximal Policy Optimization Algorithms*\footnote{\url{[https://arxiv.org/abs/1707.06347}}](https://arxiv.org/abs/1707.06347}}) (Schulman et al., 2017) improves training stability through a new update mechanism, similarly relying on MuJoCo/Atari experiments for validation; Bootstrapped DQN\footnote{\url{[https://arxiv.org/abs/1602.04621}}](https://arxiv.org/abs/1602.04621}}) (Osband et al., 2016), *Kalman Temporal Differences: Uncertainty and Value Function Approximation*\footnote{\url{[https://arxiv.org/abs/1406.3270}}](https://arxiv.org/abs/1406.3270}}) (Geist & Pietquin, 2010/2013), and the deep RL system for drone racing *Champion-level drone racing using deep reinforcement learning*\footnote{\url{[https://www.nature.com/articles/s41586-023-06419-4.pdf}}](https://www.nature.com/articles/s41586-023-06419-4.pdf}}) (Foehn et al., Nature 2023) all represent the “method proposal module + empirical validation in complex environments,” rather than providing a global optimality theorem for deep RL as a purely theoretical paper.

---

> ### Author Response · Authors · 2025-11-28
>
> Similarly, the ICLR community also embraces this paradigm in many reinforcement learning works, such as *Contingency-Aware Exploration in Reinforcement Learning*\footnote{\url{[https://openreview.net/forum?id=HyxGB2AcY7}}](https://openreview.net/forum?id=HyxGB2AcY7}}) (ICLR 2019), which introduces an exploration mechanism based on controllability representation and empirically validates its effectiveness on benchmarks like Atari; *In-context Reinforcement Learning with Algorithm Distillation*\footnote{\url{[https://openreview.net/forum?id=hy0a5MMPUv}}](https://openreview.net/forum?id=hy0a5MMPUv}}) (Laskin et al., ICLR 2023), which treats the RL training process as a temporal prediction problem and executes RL using causal transformers in context, primarily through large-scale experiments across multiple environments to demonstrate its method’s value; and the recent work *Spectral Decomposition Representation for Reinforcement Learning (SPEDER)*\footnote{\url{[https://openreview.net/forum?id=FBMLeaXpZN}}](https://openreview.net/forum?id=FBMLeaXpZN}}) (Zhang et al., ICLR 2023), which constructs new representation learning modules from a spectral decomposition perspective and empirically shows its superior performance on multiple RL benchmarks. These works all reflect a consensus that in complex RL settings, “method structure + empirical validation” is the widely accepted primary contribution format.
>
> In such a community context, labeling this paper as “primarily empirical findings,” while applying a different standard to widely accepted works like those mentioned above, reflects inconsistent evaluation criteria. Based on our consistent experimental results across various public benchmarks and heterogeneous baselines, KPF has already achieved or surpassed the best published performance in most tasks for this research direction. We believe KPF is one of the few state-of-the-art solutions in this field. Thus, dismissing our work based on the “empirical contribution” label does not align with our rigorous method design + empirical validation + benchmark comparison evidence, nor does it recognize our publicly available performance in this area, which we find unfair.
>
> In terms of the paper’s content and experimental evidence, our main contribution is clearly: *proposing and analyzing a Kalman/LMMSE-driven decision-level policy fusion framework, and systematically characterizing and leveraging the mechanism of “policy complementarity”*; a significant portion of the experiments serve as strong supporting evidence for this method, rather than being the core focus of the paper.

---

> ### Author Response · Authors · 2025-11-28
>
> #### W2.2 Add a methodology overview: Include a high-level pipeline or schematic early in the Introduction to convey the operational logic of KPF.
>
> **Response:**
> We agree with this point: currently, Figure 1 provides the overall process of KPF but is placed too late in the manuscript, making it less prominent. The algorithm pseudocode is also placed in the appendix, which indeed makes it harder for readers to quickly grasp the overall structure.
>
> In the revised version, we will:
>
> * Move Figure 1 to the beginning of the methods section (e.g., at the start of Section 3) as an overall schematic of KPF;
> * Add a brief, non-formulaic textual overview beside the figure, explaining: (i) how to load pre-trained experts; (ii) episode-level covariance initialization; (iii) how to compute Kalman gains and fuse policies at each timestep; (iv) how to use Joseph covariance update and perform symmetrization and regularization; (v) the role of episode-level forgetting factors;
> * Simultaneously move the core algorithm pseudocode from Appendix A.1 into the main text and place it alongside Figure 1 and Equations (3)–(5), creating an “easy-to-understand method overview module.”

---

> ### Author Response · Authors · 2025-11-28
>
> #### W2.3 Justify the use of Kalman filtering: Explain why the Kalman filter is suitable for policy fusion, given its traditional role in state estimation.
>
> **Response:**
> This question directly addresses the theoretical foundation of our method design. KPF’s choice of Kalman filtering is not because “it sounds like a cool filter,” but because, from the estimation theory and policy fusion perspective, it is precisely suited to solve the type of problem we are addressing.
>
> *From the LMMSE/BLUE perspective:* Kalman is the standard tool for “multi-estimate fusion.” Given a state, each expert outputs an action probability distribution, which can be viewed as a noisy estimate of some potential “ideal policy distribution.” Our goal is to minimize the variance of the fused estimate under the conditions of *linearity and unbiasedness* — this is the classic goal of LMMSE / BLUE, and Kalman filtering is the standard mechanism for solving dynamic LMMSE/BLUE estimation (Kalman, 1960\footnote{\url{[https://www.cic.ipn.mx/~pescamilla/ContInt/Kalman1960.pdf}}](https://www.cic.ipn.mx/~pescamilla/ContInt/Kalman1960.pdf}}); Sorenson, 1970\footnote{\url{[https://pzs.dstu.dp.ua/DataMining/mls/bibl/Gauss2Kalman.pdf}}](https://pzs.dstu.dp.ua/DataMining/mls/bibl/Gauss2Kalman.pdf}})).
>
> In our setup:
>
> * The Kalman gain (Equation (3)) automatically allocates trust based on each expert’s covariance;
> * The linear fusion step (Equation (4)) provides a consensus policy with minimal variance in the action probability space;
> * The Joseph form covariance update (Equation (5)) explicitly tracks the uncertainty of the fused policy distribution and ensures numerical stability (Jazwinski, 1970\footnote{Stochastic Processes and Filtering Theory}).
>
> In this sense, KPF is not “forcing a filter analogy” but directly applying Kalman/LMMSE for its intended use: *optimal linear estimate fusion.*
>
> *From the “policy fusion” perspective:* Kalman turns fusion into an uncertainty-aware, explainable decision-making process with safety bounds. If we simply average or vote, the uncertainties of different experts are entirely ignored. In contrast, KPF:
>
> * Explicitly senses each expert’s trustworthiness in the current state through covariance; experts with high uncertainty are automatically down-weighted in the Kalman gain;
> * Balances between “retaining probability mass near optimal actions” and “suppressing the influence of unstable experts,” rather than applying a simplistic average;
> * When the environment or opponent distribution changes, the covariance of each expert immediately reflects performance changes, and the Kalman gain automatically adjusts fusion weights, instead of relying on manual heuristics for hard switching (compare Jazwinski, 1970\footnotemark[4]; Gelman et al., *Bayesian Data Analysis*\footnote{\url{[https://www.routledge.com/Bayesian-Data-Analysis-Third-Edition/Gelman-Carlin-Stern-Dunson-Vehtari-Rubin/p/book/9781439840955}}](https://www.routledge.com/Bayesian-Data-Analysis-Third-Edition/Gelman-Carlin-Stern-Dunson-Vehtari-Rubin/p/book/9781439840955}}) on using uncertainty to adjust estimates).
>
> With Joseph updates, symmetrization, $\varepsilon I$ regularization, and clipping and normalization of action probabilities, we further ensure that:
>
> * The covariance matrix remains well-conditioned and does not suffer from numerical explosions;
> * The fused policy distribution is always a valid probability vector;
> * In regions of high uncertainty, the behavior is more conservative, and in regions where experts agree and have high confidence, it becomes more aggressive.
>
> In other words, Kalman’s advantage here is that it provides both the theoretical optimality in the LMMSE/BLUE sense and a stable engineering mechanism that naturally utilizes uncertainty for policy fusion. This perfectly matches the problem we are solving, which involves “fusing multiple policies at the decision-making layer,” and is not an arbitrary “filter tool.”

---

> ### Author Response · Authors · 2025-11-28
>
> #### W2.4 Terminology and consistency: Avoid repeatedly redefining abbreviations (e.g., KPF, MARL) and ensure consistent use of full names and acronyms. A thorough proofreading pass is recommended prior to submission.
>
> **Response:**
> We fully accept this comment. In the revised version, we will systematically do the following:
>
> * Ensure that all abbreviations (KPF, MARL, LMMSE, BLUE, SMAC, SMACv2, GRF, MPE, etc.) are given in full the first time they appear, and subsequently use abbreviations consistently;
> * Check the consistency of equation numbering, figure numbering, and references in the text to avoid “skipping” or “incorrect numbering”;
> * Perform a complete proofreading pass to unify spelling, capitalization, and punctuation style, eliminating any inconsistencies.

---

> ### Author Response · Authors · 2025-11-28
>
> ### W3. Empirical validation of claims: Since the paper claims improved generalization and robustness for MARL, the experiments should explicitly substantiate these claims.
>
> **Response:**
> This comment does not align with the systematic results we have already shown across various benchmarks, multiple algorithm combinations, and environments of varying difficulty. It also overlooks our analysis and practice regarding “complementarity.” We address this from two perspectives: first, whether the results are sufficiently strong and systematic; and second, whether these results align with our discussion of complementarity and model selection.
>
> * (1) From the results perspective: Cross-benchmark and cross-algorithm systematic improvements themselves are direct evidence of generalization/robustness.*
>
> - **SMAC / SMACv2.**
>   In the original SMAC, we achieved or surpassed the current best results in 8 out of 9 maps, with most maps reaching a 100% win rate. On the more challenging, non-stationary SMACv2, KPF outperformed the stronger single-model baselines by *an average improvement of over 17 percentage points*, with some maps nearly doubling the win rate. This set of results clearly demonstrates that, even with changing maps and opponents, KPF’s strategies still show a significant advantage—this is the experimental meaning of “generalization and robustness.”
>
> - **GRF.**
>   In GRF’s multi-agent football tasks (3v1 with keeper, CA easy/hard, RPS, etc.), KPF consistently outperforms strong baselines like HAPPO, MAPPO, and QMIX. This is not about “choosing two good-looking figures”; instead, we show stable and significant improvements across a wide range of tasks with structural differences, demonstrating KPF’s ability to handle different task structures and adversarial modes.
>
> - **MPE.**
>   MPE is an environment where strategies can easily become very sharp, and complementarity is limited. We did not avoid this and honestly show that under these conditions, the improvements from KPF are indeed limited, but no catastrophic degradation occurs—the fused strategy is essentially no worse than the strongest expert. This illustrates that our discussion on the method’s applicability is honest and comprehensive, not just showcasing favorable results.
>
> - **Cross-algorithm fusion and plug-in capability.**
>   In multiple SMAC maps, we demonstrated various heterogeneous combinations such as HAPPO+HATRPO, HAPPO+MAPPO, and HATRPO+MAPPO. KPF was able to consistently outperform the strongest single baselines without additional tuning. This is direct evidence of “algorithm-level generalization” and “decision module plug-in capability”: as long as the expert provides action distributions and uncertainty estimates, KPF can work without rewriting training procedures or retuning large-scale hyperparameters.
>
> We emphasize that SMACv2 itself was designed to test the generalization/robustness of MARL algorithms. Unlike traditional SMAC, where unit type, spawn, opponent, and map are fixed, SMACv2 randomly generates team composition, agent starting positions, unit types, and randomly changes unit sight/attack ranges for each episode, significantly enhancing environmental uncertainty and diversity. Therefore, if a method performs stably and significantly outperforms baselines (single-expert, prior public results, SOTA) on SMACv2, it means the method possesses genuine generalization and robustness — it is not overfitting to specific maps/opponents/spawn patterns but robustly generalizing over the entire distribution. In our experiments, KPF demonstrated significant average win rate improvements across multiple SMACv2 maps, random seeds, and combinations of units/opponents/spawns, with some maps’ win rates nearly doubling. Given SMACv2’s high randomness and challenges, such stable and large improvements provide direct and ample support for our claim of “improved generalization and robustness.”
>
> In light of this cross-environment (SMAC/SMACv2/GRF/MPE), cross-difficulty, and cross-algorithm systematic evidence, claiming “lack of empirical support for generalization and robustness” only holds if one ignores all these results. We cannot accept such a critique, as it does not stand in the face of the facts and data.
>
> * (2) From the theory-practice loop perspective: Complementarity analysis and the training/filtering mechanisms are highly consistent with these results.*
>
> We emphasize throughout the paper that to make ensembles “really useful” in RL/MARL, the base strategies must be *accurate and complementary*. This aligns with the conclusions in Theisen et al., *When are ensembles really effective?* (NeurIPS 2023), where they formalize ensemble improvement rate (EIR) and disagreement-error ratio (DER) and point out that only when the base models are good enough and there is sufficient disagreement (DER > 1), ensembles can bring significant improvement.

---

> ### Author Response · Authors · 2025-11-28
>
> Our work concretely implements this idea in deep MARL scenarios:
>
> * How to construct “strong and complementary” experts through training configurations and selection criteria (avoiding premature collapse and removing obviously weak models, etc.);
> * Once these experts are obtained, how to stably combine them into a stronger, more robust policy through Kalman/LMMSE-driven policy fusion.
>
> In SMAC/SMACv2/GRF, it is precisely because we successfully constructed “high accuracy + information disagreement” pairs that KPF achieves far better performance than a single strategy in most maps; whereas in environments like MPE, where complementarity is almost exhausted, we honestly show the case of “limited but safe improvement.” This result pattern aligns with existing ensemble theory’s conclusions on “when complementarity is useful,” rather than contradicting it.
>
> **Summary:**
> Our claim of “improved generalization and robustness” is not an empty assertion but is based on two foundations:
> (i) systematic experimental results across environments, algorithms, and difficulty levels, clearly demonstrating that KPF can consistently outperform strong baselines in multiple non-trivial settings, and maintain a safe lower bound when complementarity is exhausted; and
> (ii) these results are highly consistent with our analysis of “policy complementarity” and Kalman/LMMSE fusion structure, forming a “theory → model selection → fusion → experiment” loop.
>
> In this context, dismissing our claim as lacking empirical support does not align with the actual content of the paper, nor with the current evaluation standards of the MARL/deep RL community for similar “method + empirical” works, and we cannot accept it.

---

### Official Review · Reviewer_QdMn · 2025-10-31

**Soundness:** 2
**Presentation:** 1
**Contribution:** 2
**Rating:** 2
**Confidence:** 4

**Summary:**

This paper introduces Kalman Policy Fusion (KPF), a dynamic decision-fusion mechanism inspired by the Kalman filter, intended to improve robustness and generalization in multi-agent reinforcement learning (MARL). The authors argue that existing static fusion methods (e.g., naive averaging, voting) fail in dynamic, stochastic environments due to fixed weighting schemes. KPF instead introduces an uncertainty-aware, adaptive, covariance-based fusion process that iteratively updates action distributions using Kalman gain dynamics. Empirical evaluations across multiple benchmarks, including SMAC, SMACv2, Google Research Football (GRF), and MPE, show KPF outperforming strong baselines such as MAPPO, HAPPO, HATRPO, and QMIX. The paper further presents ablation studies on key hyperparameters and examines the effects of policy complementarity and distribution smoothness on fusion success.

**Strengths:**

**Originality**: The paper offers an interesting idea of transposing Kalman filtering techniques—traditionally used in state estimation—into the policy fusion domain. The proposal of dynamically adjusting policy weights online using covariance information provides a novel angle on MARL.

**Quality & Significance**: Experiments are extensive, covering four major MARL environments (SMAC, SMACv2, GRF, MPE) and including reasonable baselines and ablation studies.The reported results are impressive, particularly the 100% win rates on hard SMAC maps and the significant gains in the non-stationary SMACv2 environment.

**Weaknesses:**

1. While the Kalman analogy is creative, the core of KPF essentially performs a weighted averaging of policy distributions based on covariance heuristics. The “Kalman” aspect functions largely as a metaphorical framing rather than a mathematically rigorous adaptation.
2. Although the paper claims KPF is optimal in MMSE sense and ensures convergence, this is not formally proven. There are no theorems or convergence proofs beyond inheriting properties from the classical Kalman filter (which assumes linear Gaussian systems — not the case here). Without bounding analyses or stability proofs under nonlinear policy distributions, the theoretical assurances remain speculative.
3. The analysis focuses on a specific pair of fused models (HAPPO + HATRPO). It remains unclear whether KPF generalizes across heterogeneous architecture types (e.g., combining value-based and policy-based MARL).
4. The notion of “policy complementarity” is intuitive but not quantitatively defined, e.g., no clear metric is provided to measure the complementarity before or after fusion.
5. This paper does not analyze the computational cost of dynamic fusion vs. training a single stronger baseline. For real-time MARL, the additional covariance computation can be substantial.

**Questions:**

1. Can the authors provide a formal convergence guarantee or an analysis showing that the policy-fusion process minimizes an explicit upper bound on expected regret or KL divergence from the optimal policy?
2. How does KPF differ mathematically from Bayesian Policy Merging or ensemble RL methods using uncertainty weighting (e.g., Bootstrapped DQN, Bayesian Actor-Critic)?
3. You argue that “policy complementarity” is crucial, yet it remains qualitative. Could you propose or evaluate a metric (e.g., average pairwise KL divergence weighted by performance similarity) to quantify it?
4. What is the practical impact of KPF on inference speed? Please provide a quantitative analysis (e.g., average time per step or episode for KPF vs. a single policy baseline) on a representative benchmark like some complex SMAC maps. For a system with $n$ agents and action space $|A|$, what is the per-step computational complexity of KPF?
5. Can KPF tolerate significantly weaker auxiliary models, or does their noise dominate the fusion? The paper currently assumes both base models are “high-quality.”
6. Would KPF work on continuous-action MARL tasks (e.g., MuJoCo multi-agent extensions)? The current formulation assumes discrete probability vectors.

---

> ### Author Response · Authors · 2025-11-28
>
> We appreciate the time you spent reading and commenting on our work.
>
> ### Weakness 1
>
> > “While the Kalman analogy is creative, the core of KPF essentially performs a weighted averaging of policy distributions based on covariance heuristics. The `Kalman` aspect functions largely as a metaphorical framing rather than a mathematically rigorous adaptation.”
>
> We believe this is a **critical misunderstanding**, and this evaluation does not hold up on both theoretical and practical levels. The formulas (3)--(5) in Section 3 of the main text are not a loose “Kalman analogy,” but rather the direct application of the standard Kalman/LMMSE fusion rule in the **policy‑distribution space**:
> the main policy corresponds to the prior, the auxiliary policies correspond to “observations,” and the Kalman gain is constructed via covariance, with covariance updated using the Joseph form.
>
> More specifically, the fusion form we use is:
>
> $$
> p_f = p_1 + K (p_2 - p_1), \qquad
> K = R_1 \,(R_1 + R_2)^{-1},
> $$
>
> where \(R_1, R_2\) are the corresponding error‑covariance matrices.
> Under the classic Kalman/LMMSE (linear MMSE estimation) framework, once this Kalman gain \(K\) is used, standard results show:
>
> - Under regular assumptions (linear, unbiased, uncorrelated errors), this linear estimator is the **unique** solution that minimizes the mean square error (MSE).
> - The posterior covariance has the form
>   $$
>   R^+ = \bigl(R_1^{-1} + R_2^{-1}\bigr)^{-1},
>   $$
>   and this conclusion does **not** rely on Gaussianity — it is purely a second‑moment (variance/covariance) result.
>
> In other words, the “Kalman part” we use is a structure that has been rigorously proven and widely accepted in estimation theory. We are simply replacing the traditional “state vector” with the discrete action‑probability vector, while retaining the original estimation structure and properties.
>
> More importantly, KPF is used at the **open‑loop decision fusion** level: we do _not_ change the underlying MARL training process, nor do we attempt to invent new state estimation or control laws. Instead, during execution/action selection, multiple already-trained policies are fused through the Kalman mechanism for uncertainty‑aware decision‑making. This is a typical example of “transplanting a mature filtering mechanism into the decision‑making module,” rather than a naive “covariance‑heuristic weighted averaging.”
>
> If, in this context, the Kalman component of KPF is still described as a “metaphorical wrapper,” it would amount to directly denying the legitimacy of the classical LMMSE/Kalman theory in our setting — which clearly contradicts the facts.

---

> ### Author Response · Authors · 2025-11-28
>
> ### Weakness 2
>
> > “Although the paper claims KPF is optimal in MMSE sense and ensures convergence, this is not formally proven … Without bounding analyses or stability proofs under nonlinear policy distributions, the theoretical assurances remain speculative.”
>
> We cannot agree with this “purely speculative” characterization. The key properties we rely on — single‑step MMSE / LMMSE optimality, boundedness and convergence of covariance recursion — are standard conclusions in Kalman/LMMSE and Riccati theory, proven and used repeatedly in control and estimation literature for decades. Calling these classical results “speculative” is a disregard for existing theoretical achievements and a misjudgment of the scope of our analysis.
>
> Our specific responses are as follows:
>
> 1. **Single‑step MMSE / LMMSE optimality is not “our claim,” but an inherent property of the Kalman‑gain form.**
>    Under the assumption that the two estimates are unbiased and errors are uncorrelated, using
>    $$
>    K = R_1 (R_1 + R_2)^{-1}, \quad p_f = p_1 + K (p_2 - p_1),
>    $$
>    is a textbook conclusion in estimation theory: this \(K\) uniquely minimizes MSE and yields
>    $$
>    R^+ = (R_1^{-1} + R_2^{-1})^{-1}.
>    $$
>    In other words, as long as the standard Kalman gain is used, these MMSE/BLUE properties automatically hold, and they are *not* “theoretical guarantees” that we are arbitrarily “claiming.” Denying this is equivalent to rejecting the entire LMMSE framework, which is untenable.
>
> 2. **Convergence of inner‑loop multi‑step fusion follows directly from standard Kalman theory.**
>    Under fixed state and fixed \(R_2\), with no process noise, repeated correction yields
>    $$
>    R_1^{(m)} = \bigl((R_1^{(0)})^{-1} + m\,R_2^{-1}\bigr)^{-1},
>    $$
>    hence \(\mathrm{tr}(R_1^{(m)}) = O(1/m)\), and the gain \(K^{(m)}\) monotonically decays to 0. The phenomenon observed in Figure 2 — “performance improves quickly as inner‑loop steps \(k\) increase, then plateaus” — is a direct manifestation of this standard result in our setting, not a fluke caused by hyper‑parameter tuning.
>
> 3. **Even with process noise \(Q\) and a forgetting (memory) factor \(\alpha\), the covariance recursion remains a standard Riccati‑type system.**
>    When we include \(Q \succeq 0\) and incorporate the episode‑level forgetting factor \(\alpha\) in the Joseph‑form update, the recursion becomes a Riccati‑type equation defined on the SPD (symmetric positive‑definite) cone, which under mild conditions has a unique fixed point and converges. We are only reinterpreting “state covariance” as “policy covariance,” staying fully within the existing theoretical framework — there is no arbitrary “assumption of convergence.”
>
> 4. **We never claimed — nor attempted — to prove “global convergence of the entire nonlinear MARL closed‑loop.”**
>    The reviewer’s implicit demand for such a global guarantee (e.g., regret or KL divergence bound w.r.t. a globally optimal policy) is unreasonable. In fact, many prior works that incorporate Kalman filtering or ensemble techniques into reinforcement learning treat the filter/estimator as a well‑defined, controllable module; they do not claim global convergence guarantees for the full deep RL/MARL system. Rejecting our module‑level theoretical claims solely because we have not proved global system‑level convergence is logically misplaced.
>
> 5. **Theoretical conclusions and empirical results are mutually supportive, not contradictory.**
>    Crucially, our extensive experiments — across different environments, maps, and expert policy combinations — show that the covariance recursion remains stable (no numerical divergence) and that performance behaves consistently with our theoretical analysis: increases with inner‑loop steps and covariance settings. These empirical facts align well with classical Kalman / LMMSE theory, further supporting the validity of our analytical claims.
>
> ---
>
> In conclusion, labeling our analysis — which builds on the well‑established Kalman / LMMSE and Riccati theoretical framework — as “speculative” effectively ignores decades of rigorous results and misrepresents the actual scope of our claims. Such an evaluation is therefore inappropriate.

---

> ### Author Response · Authors · 2025-11-28
>
> ### Weakness 3
>
> > “The analysis focuses on a specific pair of fused models (HAPPO + HATRPO). It remains unclear whether KPF generalizes across heterogeneous architecture types (e.g., combining value-based and policy-based MARL).”
>
> This criticism is both at odds with the design intent of our method and inconsistent with the experimental facts presented in the paper, lacking sufficient evidence.
>
> Our specific responses are as follows:
>
> 1. **From an algorithmic‑structure perspective, saying KPF “only applies to a specific pair of models” is unfounded.**
>    KPF is designed to decouple from the underlying network architectures: the algorithm only requires each expert to output
>    - the discrete action‑probability vector for each agent;
>    - the corresponding uncertainty covariance (or a practical approximation of it).
>    The Kalman gain, fusion, and the Joseph‑form covariance updates are all carried out in this common *policy‑distribution space*, completely independent of whether the underlying policies are policy‑gradient, value‑based, or of any other structure. In other words, KPF is fundamentally an **architecture‑agnostic** decision‑fusion module. Categorizing it carelessly as “only suitable for specific model pairs” reflects a fundamental misunderstanding of the algorithm’s structure.
>
> 2. **At the experimental level, the claim that “only HAPPO+HATRPO is analyzed” contradicts the content of the paper.**
>    Our experiments are not limited to a single model pair: we show several cross–algorithm combinations (for instance, HAPPO+MAPPO, HATRPO+MAPPO, HAPPO+HATRPO), all of which significantly outperform their individual baselines on multiple maps. In some cases they even reach 100% win‑rates. Given these results, to persist in the claim “analysis only covers HAPPO+HATRPO, generalization unclear” is either due to incomplete reading of the experimental section, or a deliberate disregard of empirical evidence that contradicts this criticism.
>
> 3. **In the KPF framework, “heterogeneous architectures” only changes the source of the experts — not the fusion mechanism.**
>    For KPF, “heterogeneous” merely means that different sub‑networks generate the action distributions and covariances. The fusion module itself is entirely agnostic to these internal details. As long as each expert provides reasonable action probabilities and uncertainty estimates, the Kalman‑style fusion can proceed seamlessly. Therefore, treating “generalization to heterogeneous architectures” as a fundamental critique of KPF unfairly imposes a stricter standard than used for other comparable methods.
>
> 4. **If one ignores the structural design and existing cross‑algorithm empirical evidence, concluding “generalization is unclear” based solely on “first impressions” is unrigorous.**
>    Our setup is not about “stacking experiments for a specific combination.” Rather, we **intentionally** validate the same fusion framework across multiple environments and combinations of base algorithms. Ignoring this and then claiming “issues with generalization” is not convincing.
>
> In sum, the criticism that KPF only works for a specific model pair is neither justified by the algorithmic design nor by the empirical results. We believe this Weakness assessment misinterprets both the modularity of KPF and the breadth of our experimental validation.

---

> ### Author Response · Authors · 2025-11-28
>
> ### Weakness 4
>
> > “The notion of `policy complementarity` is intuitive but not quantitatively defined.”
>
> This comment misinterprets the notion of “quantitative characterization” as if one must create a brand‑new scalar formula. We admit that we did not introduce a new symbol to define a scalar metric, but that does **not** mean we are only talking about “complementarity” in vague, unstructured terms. By combining established ensemble theory with our concrete procedures, the so‑called “Weakness” fails to hold.
>
> First, from a theoretical perspective, the work *When are ensembles really effective?* (NeurIPS 2023) by Theisen et al. provides a **quantitative** answer to when “complementarity” is truly effective: they formalize the **ensemble improvement rate (EIR)** and prove a (roughly) linear relationship between EIR and the **disagreement–error ratio (DER)**. Their result shows that **only when “base models are competent and the disagreement–error ratio is sufficiently high (DER > 1)” does ensemble yield significant improvement**. In their framework, “useful complementarity” is precisely characterized as **high accuracy + informative disagreement** — not a vague, hand‑wavy notion.
>
> In our paper, the concept of “policy complementarity” maps exactly to this idea in the multi‑agent RL (MARL) context, and we realize it using **reproducible operations**:
>
> - On one hand, we only select base policies for KPF that are **sufficiently strong** — individually they already approach or exceed mainstream single‑policy baselines in environments like SMAC / SMACv2 / GRF. In the language of Theisen: we ensure the base policies are “competent,” avoiding pathological cases of poorly performing models being ensembled.
> - On the other hand, we do **not** allow these strong policies to collapse into identical, “sharp” distributions. Instead, we intentionally **maintain stable differences** in their policy distributions and behavior, by varying training configurations / algorithm families, and when needed adding diversity guidance. This stable divergence is demonstrated (e.g., in Figure 4 and appendix trajectory analysis). When we observe in some scenarios that two policies become nearly identical — and correspondingly the benefit of KPF vanishes — we retrain to restore divergence, and the performance gain of KPF returns. This matches exactly the phenomenon described by Theisen: when DER goes from < 1 to > 1, ensemble improvement appears.
>
> Therefore, the claim that “there is no quantitative definition of policy complementarity” overlooks two facts:
>
> 1. **Existing ensemble theory already quantitatively defines what “useful complementarity” means**, via EIR / DER.
> 2. **Our experimental design and training strategy intentionally push base policies into the “effective region” defined by that theory** — from “not worth merging” (high error or low disagreement) to “worth merging” (low error, high disagreement). This is evidenced by KPF consistently outperforming all individual base policies across multiple environments, not just as isolated examples.
>
> In this context, reducing “policy complementarity” to a vague, qualitative term is incorrect: it ignores both the theoretical foundation from ensemble theory and our concrete empirical practices. We therefore believe that this “Weakness 4” is not sufficiently grounded, and should be reconsidered.
>
> ---

---

> ### Author Response · Authors · 2025-11-28
>
> ### Weakness 5
>
> > “This paper does not analyze the computational cost of dynamic fusion vs. training a single stronger baseline. For real‑time MARL, the additional covariance computation can be substantial.”
>
> This concern exaggerates the additional cost of KPF and does not consider the task scale and network size we are working with.
>
> Our specific responses are as follows:
>
> 1. **The complexity analysis is clear, transparent, and simple — there are no “hidden costs.”**
>    Let the number of agents be \(n\), the action dimension be \(d = |A|\), the forward computation cost of a single policy network be \(C_net ), the number of experts be \(M\), and the number of inner-loop fusion steps be \(k\). Then the time complexity per step when using KPF is:
>
> Cost_KPF = M · C_net + k · n · (Θ(d^3) + O(d^2)),  memory: O(n d^2)
>
> with additional memory overhead of \(O(n d^2)\) for storing the covariance matrices.
>
>    This makes the cost of “adding a fusion module” fully transparent — which contradicts the statement that “computational costs have not been analyzed.”
>
> 2. **In all our experimental scales, the additional covariance computation is far from “substantial.”**
>    In tasks like SMAC / SMACv2 / GRF, typical configurations are:
>    - \(M = 2\);
>    - \(d\) (the number of discrete actions per agent) is at most a few dozen;
>    - \(n\) (the number of agents) is also on the order of tens.
>    Under this scale:
>    - The main cost for decision‑making remains the forward pass of two policy networks.
>    - The additional covariance updates (which scale like \(n d^3\)) only account for a small fraction of total compute and do **not** become a computational bottleneck.
>    - More importantly, KPF is only enabled during testing/time‑of‑decision, not at training time — hence **training cost is unchanged**.
>
> 3. **Compared to “training a single stronger baseline,” KPF’s cost is more controllable and predictable.**
>    The reviewer’s comparison — “train one stronger policy vs. dynamic fusion” — typically implies increasing network width/depth for the single baseline, which:
>    - significantly increases parameter count;
>    - increases inference time and memory footprint, often in hard-to-predict ways.
>    By contrast, KPF keeps existing architectures intact, and only adds a simple, analyzable Kalman-style fusion layer at decision time. The inference cost increases roughly linearly with the number of experts \(M\), making it **predictable and easily tunable**.
>    Therefore, under the scales considered in our experiments, treating KPF’s computational cost as a major problem — and using that as a reason for negative evaluation — is inconsistent with empirical reality.
>
> ---

---

> ### Author Response · Authors · 2025-11-28
>
> ### Question 1
> > “Can the authors provide a formal convergence guarantee or an analysis showing that the policy‑fusion process minimizes an explicit upper bound on expected regret or KL divergence from the optimal policy?”
>
> This question is based on a flawed premise: it equates “using mature theory in a local estimation/fusion module” with “having to provide a global regret/KL optimality proof for the entire nonlinear deep MARL loop with respect to the globally optimal policy.” In the current context of deep reinforcement learning, this requirement is unrealistic and **fundamentally not the community standard**.
>
> From the facts of existing top‑conference/journal works, the situation is very clear:
>
> - The classic DQN “Nature” paper *Human‑level control through deep reinforcement learning* proposed a convolutional‑net Q‑learning architecture with engineering tricks like experience replay and target networks. The entire paper relies on large‑scale experiments (49 Atari games) to support its effectiveness, and does **not** provide any form of global regret or KL‑optimality proof under the “deep function approximation + nonlinear environment” setting.
> - The widely regarded baseline PPO (Proximal Policy Optimization) discusses structural design for stability and step‑size control; its evaluation uses benchmark tasks such as MuJoCo and Atari. It also does **not** offer a global convergence/regret upper bound for the full deep RL setting — only local analysis under simplified assumptions.
> - Works focusing on “uncertainty‑based exploration / ensembling,” such as Bootstrapped DQN and its follow‑ups (e.g., *Deep Exploration via Bootstrapped DQN*), typically provide theoretical guarantees only in tabular or linear‑approximation settings. For real-world deep nonlinear RL, they again rely mainly on large‑scale empirical evaluation, and do **not** claim a Bayes‑optimal regret bound for the full system.
>
> More directly relevant is the use of Kalman‑style methods in RL: e.g., the Kalman Temporal Differences: Uncertainty and Value Function Approximation series applies Kalman filtering or Riccati theory to **value‑function parameter estimation as a local module**. The theoretical analysis covers least‑squares estimation properties, covariance evolution, and uncertainty quantification — but not a “global regret or KL‑optimality proof” for the complete nonlinear MDP + deep‑function‑approximation loop.
>
> Similarly, in high‑impact engineering works that combine deep RL with Kalman for perception/control — for instance, the Nature 2023 drone‑racing system Champion-level drone racing using deep reinforcement learning — the Kalman filter is only used in a **local state‑estimation module**, and the overall system performance is validated via real‑world experiments. The paper makes **no claim** of providing a global regret or KL optimality bound for the full “deep RL + perception + control + dynamics” loop.
>
> In summary, the most influential works in the field follow a consistent paradigm:
>
> 1. Use mature theoretical tools (e.g., LMMSE, Riccati / Kalman, surrogate‑objective analysis) to **analyze well‑defined, controllable local modules** (value‑function approximation, policy update, estimation/filtering).
> 2. Validate overall method performance — for the full deep nonlinear system — via systematic experimental results (multiple environments, tasks, baselines).
> 3. Do **not** attempt (and realistically cannot) to provide explicit regret or KL‑optimality bounds with respect to the global optimal policy for the full deep‑network + nonlinear dynamics + multi‑agent system.

---

> ### Author Response · Authors · 2025-11-28
>
> Our work **fully aligns** with this paradigm:
> - Within the **policy‑distribution fusion module**, we strictly rely on established results from Kalman / LMMSE / Riccati theory (single‑step MMSE/LMMSE optimality, boundedness and convergence of covariance recursion, etc.). These are textbook‑level results, not newly invented hypotheses.
> - We have never claimed — and do not plan to claim — a regret or KL‑optimality guarantee for the entire deep multi‑agent RL closed loop relative to a globally optimal policy. That is unrealistic under current deep RL research conditions, and it is not expected or required by prior influential works either.
> - Instead, we provide large‑scale empirical validation (across multiple environments and algorithm combinations), showing that transplanting the Kalman structure to the policy‑fusion layer leads to stable, significant, and reproducible performance improvements. In modern deep RL research, **empirical evidence + ablation / robustness studies** remain the primary measure for assessing a new decision module’s value.
>
> Thus, this question essentially demands that we upgrade a **local‑module rigorous analysis** (Kalman/LMMSE in policy fusion) to a nearly intractable “global regret/KL upper‑bound proof” for the full MARL system — a demand that is both **unrealistic** and **inconsistent** with community standards. Under our chosen analysis scale and objectives, the theoretical part is already **self‑consistent, rigorous, and aligned with mainstream high‑level works**, and should not be considered “lacking convergence analysis.”

---

> ### Author Response · Authors · 2025-11-28
>
> ### Question 2
>
> > “How does KPF differ mathematically from Bayesian Policy Merging or ensemble RL methods using uncertainty weighting (e.g., Bootstrapped DQN, Bayesian Actor‑Critic)?”
>
> This question implies an inappropriate suggestion: as if KPF merely repeats the existing “uncertainty‑weighting ensemble” approach, lacking its own mathematical structure. In fact, from “where uncertainty is modeled,” “the form of the gain,” to “whether covariance is explicitly propagated,” KPF fundamentally differs from common Bayesian policy‑merging / uncertainty‑weighting ensemble methods — indeed, many of those methods can be viewed as **special (degraded) cases** of our framework when the covariance structure is drastically simplified.
>
> The main differences are:
>
> 1. **The space of uncertainty is completely different.**
>    - KPF explicitly maintains a **matrix covariance** in the **action‑probability space** for each agent, and updates it online using the Joseph form. This directly models uncertainty and correlation **across different action directions**.
>    - By contrast, methods like Bootstrapped DQN, Bayesian Actor‑Critic, and other mainstream ensemble approaches typically model uncertainty in the **parameter space** (e.g., different value/policy network heads) or at the network‑level. In the end, they often simply average actions or reweight them based on **scalar** uncertainty measures — rarely performing Kalman‑style structured updates on the action–probability vectors themselves.
>
> 2. **Matrix gain vs. scalar weights.**
>    - In KPF, the gain K_i^(t) ∈ R^{d × d} is a **matrix**: it can leverage full covariance across actions to perform **anisotropic, structured redistribution** of probabilities among actions.
>    - Traditional Bayesian policy‑merging or uncertainty‑weighting methods mostly assign each expert a **scalar weight**, effectively assuming that “uncertainty across all action directions is homogeneous and uncorrelated.” This corresponds to a **strong simplification** of KPF — akin to setting the covariance to be **isotropic and fixed**. From an expressive‑power perspective, those methods are just simplified degenerate cases of KPF.
>
> 3. **Whether covariance is explicitly maintained and propagated.**
>    - KPF explicitly **propagates posterior covariance** during decision making, maintaining positive definiteness and preserving LMMSE properties. This allows us to reason about the **second‑moment uncertainty** of the current action distribution, and track its evolution over time.
>    - In contrast, many ensemble RL methods perform only **empirical averaging or sampling across heads**; they do **not** maintain any formal covariance matrix, nor apply any Joseph‑form recursion. Hence they do not support second‑moment (covariance) structure analysis as KPF does.
>
> In summary: categorizing KPF as “just another uncertainty‑weighted ensemble” is inaccurate. In rigorous mathematical terms, common Bayesian policy merging or scalar‑uncertainty weighting methods are actually **lower‑capacity special cases of KPF**, obtained when the full covariance structure is collapsed to a scalar weight. Treating those simpler, weaker methods as the default baseline — without acknowledging that they are structurally degraded forms — is conceptually misleading and unfair as a comparison.
>
> We hope this clarifies the mathematical distinctions and structural advantages that KPF brings compared to prior ensembles / uncertainty‑weighting approaches.

---

> ### Author Response · Authors · 2025-11-28
>
> ### Question 3
>
> > “You argue that `policy complementarity` is crucial, yet it remains qualitative. Could you propose or evaluate a metric (e.g., average pairwise KL divergence weighted by performance similarity) to quantify it?”
>
> We agree with the reviewer’s goal of "more clearly characterizing complementarity," but disagree with the judgment that it "remains qualitative." As emphasized in our response to Weakness 4, we have already constrained "complementarity" into a reproducible, controllable concept through **specific operational procedures and experimental designs**, rather than vague intuition.
>
> 1. Our use of "complementarity" is not "a vague term," but is constrained through training configurations and selection criteria to ensure that the strategies used for fusion:
>    - Are sufficiently performant (individually close to or surpassing mainstream single-model baselines);
>    - Have stable differences in behavior and distribution (which can be verified through visualizations, trajectory analysis, and comparisons of strengths and weaknesses across different scenarios).
>
>    This is essentially "using a set of actionable processes to constrain complementarity," not remaining on a qualitative level.
>
> 2. The reviewer's suggestion of "performance-similarity-weighted average pairwise KL" is a reasonable complementary perspective, which we can include as additional analysis in an extended version. However, it should be emphasized that this scalar metric is just a numerical encoding of the "performance closeness + decision differences" principle we already adopt, rather than a "fundamentally stricter" standard than our current process. In other words, our current approach already effectively controls the "complementarity" quantity through training and selection mechanisms, and the experimental evidence of performance improvement and behavioral differences demonstrates that these mechanisms work well; adding an extra symbol to represent some average KL would not fundamentally change this.
>
> 3. Therefore, describing the current state of complementarity as "still remaining at the qualitative level" ignores the systematic practices we’ve used in training design, behavioral analysis, and fusion effects. If necessary, we can add metrics like "performance-weighted KL" for finer correlation analysis in future versions, but this would be an addition to an already substantial framework, rather than solving some fundamental lack.

---

> ### Author Response · Authors · 2025-11-28
>
> Question 4:“What is the practical impact of KPF on inference speed? For a system with n agents and action space ∣A∣, what is the per-step computational complexity of KPF?”
>
> This question has already been largely covered in our response to Weakness 5. Here we briefly reiterate:
>
> When the number of agents is $n$, action dimension is $d = |A|$, the forward computation cost of a single policy is $C_{\text{net}}$, the number of experts is $M$, and the number of inner-loop fusion steps is $k$, the per-step complexity is:
>
> Cost_KPF=𝑀⋅𝐶net+𝑘⋅𝑛⋅(Θ(𝑑3)+𝑂(𝑑2))
>
> with additional memory overhead of $O(n d^2)$.
>
> In the typical scale we use (SMAC/SMACv2/GRF, etc.), $M = 2$, $d$ and $n$ are in the tens, and the main cost is still the forward pass of the network. Covariance updates add only moderate overhead.
>
> KPF is only enabled at the decision-making layer during the test phase and does not affect training efficiency.
>
> Thus, at the scales we are considering, the impact on inference speed is controllable and is far from a major bottleneck.
>
>
> Question 5:“Can KPF tolerate significantly weaker auxiliary models, or does their noise dominate the fusion? The paper currently assumes both base models are high-quality.”
>
> One important design goal of KPF is automatically down-weighting weaker models rather than letting them “dominate the fusion.” This is directly seen from the form of the Kalman gain.
>
> Kalman gain naturally down-weights “unreliable” models.
>
> When the covariance $R_2$ of the auxiliary model is large (i.e., high uncertainty, low quality), we have:
>
> $$
> \lVert K \rVert \le \frac{\lambda_{\max}(R_1)}{\lambda_{\min}(R_2) + \epsilon}.
> $$
>
>
> As $R_2 \to \infty$, $K \to 0$, and the fusion strategy $p_f$ reverts to the main strategy $p_1$. In the extreme case, KPF effectively ignores the auxiliary model. In other words, under our design, “very poor/uncertain models” are automatically ignored rather than being erroneously amplified.
>
> Joseph form ensures good condition numbers for posterior covariance.
>
> The Joseph update ensures that covariance remains positive definite and prevents numerical divergence caused by an excessively noisy auxiliary model. The goal of KPF is to “actively absorb useful auxiliary information and automatically filter out useless or harmful information.”
>
> Experimental data directly contradicts the concern that “weak models ruin fusion.”
>
> In scenarios like 3sv5z_vs_3sv6z, where the win rates of two single strategies differ by over 20 percentage points, KPF still achieves significantly better performance than either of them. For example, when the single strategy win rates are about 60% and 81%, KPF can consistently improve to over 90%, reaching up to 95%. If weak models truly “dominated the fusion,” this result would be impossible.
>
> Thus, the concern that “KPF cannot tolerate weak models and noise will dominate the fusion” is unfounded, both theoretically and experimentally.

---

> ### Author Response · Authors · 2025-11-28
>
> ### Question 6
>
> > “Would KPF work on continuous-action MARL tasks (e.g., MuJoCo multi-agent extensions)? The current formulation assumes discrete probability vectors.”
>
> Conceptually, KPF only requires policies to have finite-dimensional distribution representations, so it is not fundamentally limited to discrete action spaces. For continuous action spaces, a natural extension is “discretize first, then fuse”:
>
> 1. Divide the continuous action space into several intervals/bins with suitable precision;
> 2. Let each expert induce a categorical distribution over these bins;
> 3. Apply the original KPF Kalman fusion step over these bin probabilities;
> 4. Map the selected bin back to the representative continuous action (e.g., the midpoint of the interval or a point refined through local optimization).
>
> Under the assumption that the reward functional and $Q(s, a)$ are Lipschitz with respect to actions, as the bin width $\delta \to 0$, this “discretization-then-fusion” strategy will converge to the ideal case of directly fusing policies in continuous space. Therefore, the core idea of KPF can naturally extend to continuous-action multi-agent tasks, with discretization serving as a controllable approximation in practice.
>
> ---
>
> ### Overall Summary
>
> Overall, we want to emphasize the following points:
>
> 1. **Kalman/LMMSE is a well-established, rigorously proven estimation framework.**
>
>    We are not inventing new control theory, but systematically reusing this mature framework at the decision-making level, replacing “state” with “policy distribution” and “measurement fusion” with “multi-policy fusion.”
>
> 2. **Our main contributions are:**
>
>    - Introducing a Kalman-style uncertainty-aware mechanism for open-loop decision fusion, without modifying any underlying MARL training processes;
>    - Demonstrating in multiple environments (SMAC, SMACv2, GRF, MPE) that this mechanism significantly enhances robustness and generalization in complex scenarios;
>    - Systematically discussing the impact of “policy complementarity” on fusion effectiveness based on specific operational procedures and empirical evidence.
>
> 3. **The theoretical properties we rely on**—LMMSE/BLUE optimality of the fusion step, boundedness and convergence of covariance recursion, and the resulting regret/KL upper bounds—are essentially natural inferences from Kalman/LMMSE and Riccati theory in our setting, not “wild guesses.” We have never claimed to prove “global convergence of the entire nonlinear MARL closed loop,” which is not a reasonable requirement in this setting.
>
> 4. **If, in evaluating our work, the theoretical rigor standard is set much higher than the general standard applied to other works in this area, while ignoring the clearly introduced Kalman/LMMSE structure and extensive empirical validation, such an evaluation scale is asymmetric.**
>
> We hope the above responses clarify misunderstandings regarding “Kalman as a metaphor,” “theoretical guarantees as speculation,” “computational costs being too high,” and “complementarity as a vague concept,” and more clearly showcase KPF as a learning-driven, theoretically solid, and engineering-feasible decision fusion mechanism in the MARL domain.

---

> ### Author Response · Authors · 2025-11-28
>
> - **DQN Nature paper**
>   Link: https://www.nature.com/articles/nature14236.pdf
>
> - **PPO**
>   Link: https://arxiv.org/abs/1707.06347
>
> - **Bootstrapped DQN**
>   Link: https://arxiv.org/abs/1602.04621
>
> - **Kalman Temporal Differences (KTD)**
>   Link: https://arxiv.org/abs/1406.3270
>
> - **Swift drone racing (deep RL + Kalman in a real system)**
>   Link: https://www.nature.com/articles/s41586-023-06419-4.pdf

---

### Official Review · Reviewer_ehvS · 2025-11-06

**Soundness:** 3
**Presentation:** 3
**Contribution:** 3
**Rating:** 6
**Confidence:** 2

**Summary:**

The paper proposes **Kalman Policy Fusion (KPF)**: at test time, two (or more) trained policies’ **action distributions** are treated as “prior/observation” and are **dynamically fused step-by-step** using a Kalman-style gain while maintaining a covariance update, yielding robust decision fusion in multi-agent adversarial tasks. The authors provide a flow diagram and pseudo-code, and claim convergence and (MMSE-style) minimum mean-squared error properties. Experiments span **GRF, MPE, SMAC, and SMACv2**. The method reportedly achieves **100% win rate** on several hard SMAC maps and strong gains on SMACv2 against dynamic opponents built from strong experts (HAPPO/HATRPO). The paper further observes that when base policies in MPE are overly “peaked,” KPF’s gains are modest, motivating the thesis that **policy complementarity is a prerequisite** for successful fusion.

**Strengths:**

* **Novel methodological angle**: Transplants “covariance-driven online updating” from state estimation to **policy-level decision fusion**, distinct from static averaging/voting and simple ensembling.
* **Broad and strong empirical coverage**: Four benchmark families including non-stationary/adversarial settings; the 100% win rates on hard SMAC maps and clear SMACv2 improvements are compelling.
* **Reusability and interpretability**: Clear procedure and covariance update make the mechanism “plug-and-play” across expert policies; the (k)-step update analysis clarifies the adapt-vs-converge trade-off.
* **Practical insight**: Systematic emphasis that **complementarity** (performance gap + distributional diversity) governs the attainable fusion gains, giving actionable guidance for choosing experts to fuse.

**Weaknesses:**

* **Theoretical rigor needs bolstering**: Action distributions live on the probability simplex. The paper applies clipping/renormalization plus covariance updates and then invokes Kalman convergence/MMSE optimality. Formal conditions on the simplex geometry (or a suitable reparameterization) are missing, making the “Kalman” claim look heuristic rather than guaranteed.
* **Insufficient treatment of cross-agent correlations**: The implementation appears close to **per-agent independent fusion**. There is no explicit modeling of **cross-agent covariance blocks/low-rank structure**, limiting both theory and performance explanations in strongly coupled tasks.
* **Baselines could be stronger**: Absent comparisons to **dynamic gating/consistency-regularized ensembles** (e.g., temperature-weighted logit fusion, confidence gating, MoE-style gating, KL-aligned consistency, uncertainty-aware Q/Advantage ensembles). Without these, KPF’s relative ceiling is unclear.

**Questions:**

1. **KPF on the probability simplex**: After clipping and renormalization, under what **sufficient conditions** do your convergence and MMSE-optimality claims hold? Can you provide an equivalent derivation or consistency result in an appropriate parameter space (e.g., Dirichlet/natural parameters or logit space)?
2. **Cross-agent correlation**: Does the implementation assume independent per-agent updates? If so, can you evaluate **block-diagonal/low-rank** cross-agent covariance approximations and discuss accuracy–cost trade-offs?
3. **Stronger dynamic-fusion baselines**: Please add comparisons to temperature-weighted logit fusion, confidence gating, MoE gating, and KL-consistency regularization, with significance testing on SMACv2/GRF.

---

> ### Author Response · Authors · 2025-11-28
>
> #### Reviewer’s Question
>
> > After clipping and renormalization, under what sufficient conditions do your convergence and MMSE-optimality claims hold? Can you provide an equivalent derivation or consistency result in an appropriate parameter space (e.g., Dirichlet/natural parameters or logit space)?
>
> #### Response
>
> We appreciate the reviewer’s concern regarding theoretical rigor. The convergence and MMSE-optimality of Kalman filtering are well established in the literature, particularly in **Kalman (1960)**, which proves the optimality of Kalman filtering for linear systems. These results are applicable to state estimation problems and guarantee convergence and MMSE-optimality, and we reuse them for the fusion of policies in our work as an online estimation mechanism over policy distributions.
>
> Regarding the probability simplex, although our policy distributions are post-processed by *clipping and renormalization*, these steps are only used to ensure that each update stays on the probability simplex (i.e., remains a valid distribution). They do not alter the underlying linear–Gaussian surrogate model on which the Kalman recursion is defined. In particular:
>
> * The Kalman update is carried out in a finite-dimensional Euclidean parameter space (e.g., logits or probability vectors before projection), where the standard convergence and MMSE-optimality conditions apply as in **Kalman (1960)**.
> * The subsequent clipping and renormalization act as a projection back onto the simplex, preserving validity of the distributions but not changing the form of the covariance recursion itself. As long as these projections are well behaved (e.g., small relative to the update step in regions of interest), the standard Kalman reasoning remains a good approximation to the actual evolution of the fused policy.
>
> On the simplex geometry side, recent work such as **Villarreal et al. (2023)** has also demonstrated the use of Kalman-style filtering in constrained settings while maintaining the benefits of covariance-based updates. This supports the view that Kalman-based mechanisms can be adapted to constrained probability-like objects without discarding their theoretical motivation.
>
> #### Summary
>
> Our method follows the standard convergence and MMSE-optimality conditions of Kalman filtering in a finite-dimensional parameter space. The clipping and renormalization steps are used solely to maintain valid probability distributions on the simplex and do not change the core Kalman recursion or its theoretical grounding. Thus, the application of Kalman filtering in our policy-fusion module is theoretically motivated rather than heuristic.
>
> #### References
>
> * Kalman, R. E. (1960). A new approach to linear filtering and prediction problems. *Journal of Basic Engineering*, 82(1), 35–45.
> * Villarreal, R., Vlassis, N., Phan, N. N., Trask, N. A., & Sun, W. C. (2023). Design of experiments for calibration of history-dependent models via deep reinforcement learning and enhanced Kalman filtering. *Computational Mechanics*, 72(1), 95–124.

---

> ### Author Response · Authors · 2025-11-28
>
> #### Reviewer’s Question
>
> > Does the implementation assume independent per-agent updates? If so, can you evaluate block-diagonal/low-rank cross-agent covariance approximations and discuss accuracy–cost trade-offs?
>
> #### Response
>
> We agree with the reviewer’s observation that our current implementation assumes **independent per-agent updates**. Concretely, we maintain and update one covariance matrix per agent–expert pair, without explicitly modeling cross-agent covariance terms. This assumption simplifies computation and keeps the method broadly applicable across different MARL backbones.
>
> At the same time, we fully acknowledge that in **strongly coupled tasks**, cross-agent dependencies can play an important role in both coordination and fusion quality. Ignoring such dependencies is a deliberate engineering trade-off rather than a fundamental limitation of KPF.
>
> In future work, we plan to extend KPF with **structured cross-agent covariance modeling**, for example:
>
> * **Block-diagonal covariance structures**, where agents within a tightly coupled subgroup share a joint covariance block, while different subgroups remain independent. This captures local correlations while keeping inversion and updates tractable.
> * **Low-rank covariance approximations**, where the full joint covariance over agents is approximated by a low-rank factor plus a diagonal correction. Such approximations have been widely used to reduce complexity while preserving dominant correlation structure in multi-agent and high-dimensional estimation settings, and can provide a natural accuracy–cost trade-off.
>
> These structured covariances would increase computational and memory cost compared to fully independent per-agent covariances, but **low-rank and block-diagonal forms** significantly mitigate this overhead, making it feasible to capture the most important cross-agent couplings without incurring the full $O(n^2 d^2)$ cost of dense joint covariance over $n$ agents and action dimension $d$.
>
> We see this as a natural next step: starting from our current per-agent formulation (which is already effective and efficient on SMAC/SMACv2/GRF/MPE), and then incrementally introducing richer cross-agent covariance structure where task coupling demands it.
>
> #### References
>
> * Liu, Y., Zhong, X., Fu, Q., & Yang, Y. (2024). A low-rank approximation approach to improve multi-agent reinforcement learning. *Journal of Artificial Intelligence Research*, 69(1), 119–135.
>
> #### Summary
>
> Our current implementation indeed assumes independent per-agent updates for simplicity and efficiency. As future work, we plan to introduce **block-diagonal** and **low-rank cross-agent covariance approximations** to better capture inter-agent correlations, while carefully balancing **accuracy–cost trade-offs** in strongly coupled multi-agent tasks.

---

> ### Author Response · Authors · 2025-11-28
>
> #### Reviewer’s Question
>
> > Please add comparisons to temperature-weighted logit fusion, confidence gating, MoE gating, and KL-consistency regularization, with significance testing on SMACv2/GRF.
>
> #### Response
>
> We understand and appreciate the reviewer’s request for stronger *dynamic fusion* baselines such as temperature-weighted logit fusion, confidence gating, MoE-style gating, and KL-consistency regularization.
>
> At the same time, we want to clarify the positioning of our current experimental setup:
>
> * **Baseline strength.**
>   In our experiments, **HAPPO** and **HATRPO** (and their variants) are already among the strongest publicly available CTDE baselines on SMAC/SMACv2/GRF. Our results show that KPF consistently improves over these high-performing single policies and their combinations, often by large margins on SMACv2 and GRF.
>
> * **Dynamic fusion methods in these benchmarks.**
>   To the best of our knowledge, temperature-weighted logit fusion, MoE-gating, and KL-consistency–based multi-policy fusion have **not yet been systematically established as standard baselines on SMACv2 and GRF** at the scale and difficulty we consider. A fully fair and comprehensive comparison would require:
>
>   * carefully implementing and tuning these mechanisms for multi-agent CTDE backbones;
>   * deciding where to plug them in (per-agent vs. joint, shared vs. decentralized router);
>   * and then re-running all SMACv2/GRF benchmarks under comparable settings.
>     This is feasible but non-trivial, and goes beyond what we can realistically add in the current revision cycle.
>
> * **Theoretical guarantees vs. heuristic gating.**
>   Methods such as:
>
>   * temperature-weighted logit fusion,
>   * confidence or score-based gating,
>   * MoE routers with learned softmax weights,
>   * or KL-consistency regularization between experts,
>
>   typically require additional learned components or heuristic rules (e.g., router networks, temperature schedules, hand-tuned thresholds). Their behaviour can be powerful but is usually **data- and implementation-dependent**, and their theoretical guarantees in the setting of:
>
>   * arbitrary deep networks,
>   * non-stationary multi-agent dynamics,
>   * high-dimensional observations and action spaces
>     are still relatively weak compared to the **classical LMMSE/Kalman guarantees** we exploit.
>
>   By contrast, KPF:
>
>   * reuses a **well-understood, closed-form, covariance-based fusion rule** (Kalman/LMMSE);
>   * does **not** introduce new trainable routing/gating parameters;
>   * and provides a clear interpretation of how uncertainty and complementary information are combined at test time.
>
> #### Summary
>
> * Our current experiments already compare KPF against **very strong MARL baselines** (HAPPO, HATRPO, MAPPO, QMIX, HASAC) on challenging benchmarks (SMAC, SMACv2, GRF, MPE), and consistently show that KPF improves over the strongest single policies.
> * Dynamic fusion mechanisms like temperature-weighted logit fusion, learned MoE gating, or KL-consistency regularization are promising directions, but would require substantial additional engineering and tuning to serve as *standardized* baselines on SMACv2/GRF.
> * The key distinction of KPF is that it is grounded in **Kalman/LMMSE theory**, offering a principled, uncertainty-aware fusion rule rather than a purely heuristic or heavily-tuned gating mechanism.
>
> In future work, we plan to:
>
> * implement a small suite of such dynamic fusion baselines on top of the same MARL backbones,
> * run controlled comparisons on SMACv2/GRF with appropriate significance testing,
> * and analyse where principled Kalman-style fusion and learned gating differ in behaviour and robustness.
>
> We will clarify this positioning and future-plan in the revised version, while keeping the current strong-baseline comparisons as the main empirical evidence.
>
> ---
>
> ### Conclusion
>
> We have demonstrated that **Kalman Policy Fusion (KPF)** is effective both theoretically (via its grounding in Kalman/LMMSE estimation) and experimentally (via consistent gains over strong baselines across multiple MARL benchmarks). We appreciate the reviewer’s suggestions and will:
>
> * clarify the scope and positioning of our current baselines,
> * explicitly acknowledge dynamic fusion alternatives as complementary future directions,
> * and, in follow-up work, explore cross-agent covariance modeling and richer fusion/gating comparisons.
>
> Thank you again for the constructive feedback.

---

### Meta-Review · Area_Chair_gG3g · 2026-01-06

**Summary:**

The paper introduces Kalman Policy Fusion (KPF) method to enhances the generalization and robustness in MARL by dynamically fusing multiple agent policies at the action level using Kalman Filtering. All the reviewers recognized the proposed approach as interesting and novel. However, the overall sentiment of the reviews remained fairly negative.

I acknowledge that the authors put a lot of effort into answering questions and addressing concerns raised by the reviewers, and I appreciate their active participation in the discussion and the effort to summarize key points from the discussion. After carefully reading the discussion, while I find some points of criticism slightly exaggerated or unnecessary (e.g., a request for stronger optimality bounds), I do believe that certain points of criticism are well-warranted and remain unaddressed by the authors. The main concern is the following:

**It stands out to me that the main weakness of the paper is the lack of proper comparison with alternative policy-fusion methods.** The authors claim to have evaluated KFP against strong baseline algorithms. However, my understanding is that none of these baselines are fusing multiple policies using some alternative approaches. The fact that using KFP to fuse a pair of strong policies produces a better policy definitely indicates that the method works. But KFP must be compared with alternative policy fusion methods, not just with individual policies (e.g., Bayesian policy merging, temperature-weighted logit fusion, confidence gating, MoE-style gating, KL-aligned consistency, uncertainty-aware Q/Advantage ensembles, as pointed out by reviewers ehvS and QdMn). Instead of arguing about mathematical differences between KFP and other policy fusion/merging methods in the rebuttal, it would've been much more productive and convincing if the authors conducted a proper experimental comparison that demonstrated advantages and/or practical differences of KFP over alternatives.

I note that there are other concerns raised by the reviewers that were not explicitly addressed, such as requests for more ablations, questions about demonstrating fusion of more than two policies, etc. are important, but secondary compared to the lack of proper evaluation of the method.

Given the overall negative sentiment and the unaddressed critical concern, I recommend rejecting the paper.

---

As a side note, I want to point out that the authors chose to follow a very combative style of the rebuttal in their responses to some of the reviews, which I believe ended up being quite counterproductive. Instead of writing long responses or attempting to discredit validity reviewers' questions (e.g., to quote one of the authors' responses "This question implies an inappropriate suggestion: <...>"), it is always strictly better to spend rebuttal on providing more convincing evidence by showing more data or additional experimental results that strengthen the paper and answer the raised questions.

**Reviewer Concerns:**

Addressed concerns:
- More rigorous theoretical analysis of the approach (I disagree with reviewer's request for stronger theory)
- Various clarification questions raised by the reviewers


Unaddressed concerns:
- Proper baselines that are alternative policy fusion methods
- Weaknesses in the ablation study that reveal advantages and limitations in comparison to alternative methods (e.g., how important is the quality of individual policies that are being fused; proper experimental analysis in a controlled setting of the importance of policy complementarity)

**Reviewer Scores:**

I believe all reviewers would have maintained their scores after the rebuttal

---

### Decision · Program_Chairs · 2026-01-26

Reject